# A 2D Gabor-wavelet baseline model out-performs a 3D surface model in scene-responsive cortex

Anna Shafer-Skelton[1,2,3]*, Timothy F. Brady[1], John T. Serences[1,4]

1 Department of Psychology, University of California, San Diego, La Jolla, California, United States of America, 2 Department of Psychology, University of Pennsylvania, Philadelphia, Pennsylvania, United States of America, 3 Center for Perceptual Systems, University of Texas at Austin, Austin, Texas, United States of America, 4 Neurosciences Graduate Program, University of California, San Diego, La Jolla, California, United States of America

* annashaferskelton@utexas.edu

## Abstract

Understanding 3D representations of spatial information, particularly in naturalistic scenes, remains a significant challenge in vision science. This is largely because of conceptual difficulties in disentangling higher-level 3D information from co-occurring features and cues (e.g., the 3D shape of a scene image is necessarily defined by "low-level" spatial frequency and orientation information). Recent work has employed newer models and analysis techniques that attempt to mitigate these difficulties within a model-comparison framework. For example, one such study reported 3D-surface features were uniquely present in areas OPA, PPA, and MPA/RSC (areas typically referred to as 'scene-selective'), above and beyond a Gabor-wavelet baseline model. Here, we tested whether these findings generalized to a new stimulus set that, on average, dissociated static Gabor-wavelet baseline features from 3D scene-surface features. Surprisingly, we found evidence that a Gabor-wavelet baseline model— commonly thought of as a "low-level" or "2D" model—better fit voxel responses in areas OPA, PPA and MPA/RSC compared to a model with 3D-surface information. We highlight that this difference in results could be due to differences in the baseline conditions used across studies. These findings emphasize that much of the information in "scene-selective" regions—potentially even information about 3D surfaces— may be in the form of spatial frequency and orientation information often considered 2D or low-level. Disentangling lower-level and higher-level visual information is a continuing fundamental challenge for model-comparison approaches in visual cognition, and it motivates future work investigating which visual features could cue higher-level properties in our real-world visual experience—both within and beyond current model comparison frameworks.

provided the original author and source are credited.

**Data availability statement:** All ROI voxel responses are available at: https://osf.io/gmujq/overview?view_only=b12c5f-62c574479d909d4bc4461d249e, and all model fitting code and code for generating figures are available at: https://github.com/annashaferskelton/sceneSurfacesVsGabors.

**Funding:** This work was funded by: The National Eye Institute, www.nei.nih.gov (NEI R01EY025872 awarded to JTS and F32EY036266 awarded to AS); The National Science Foundation, www.nsf.gov (BCS-1653457 to TFB and NSF GRFP to AS); and The American Psychological Association, apa.org (APA Dissertation Research Award to AS). The funders had no role in study design, data collection and analysis, decision to publish, or preparation of the manuscript.

**Competing interests:** The authors have declared that no competing interests exist.

## Author summary

To gain a more complete picture of human visual processing, it is critical to understand the precise format of representations of naturalistic visual scenes. Recent work has approached this challenge by quantifying how much of our brain activity might be due to hypothesized characteristics of the stimuli being viewed. Here, we followed up on work finding that activity in scene-responsive regions of the brain is well predicted by information about the 3D configurations of major surfaces in viewed scenes, like walls and floors. In contrast to previous work, we found that our baseline condition—commonly thought of as "low-level" visual information—accounted for responses in these regions better than both of the 3D surface models that we tested. We highlight that this difference in results could be due to differences in the baseline conditions used across studies. However, our findings do not necessarily argue against the importance of these regions in encoding 3D surface information. Instead, they highlight the possibility that baseline models can perform well by virtue of covariation between low-level features like orientation and spatial frequency with higher-level properties like depth information. This motivates future work and/or new analysis frameworks to better characterize the interplay between hypothesized model features and the specific techniques used to quantify their relationship to neural representations.

## Introduction

A large body of evidence in vision science argues that anatomically distinct regions of cortex are selective for images of natural scenes as opposed to discrete visual objects [1,2]. In particular, early work suggested that these "scene areas" were especially sensitive to configurations of major surfaces like walls and floors that define the 3D shape of a space [2–4]. However, studying the content of this scene information is challenging because it is difficult to disambiguate representations of scene surfaces per se both from representations of more global scene properties (like "openness" or "navigability") and from low-level features (like orientation or spatial-frequency information). Three well-studied regions of cortex are fruitful targets to investigate the contributions of each of these factors (surfaces, global properties, low-level features): the occipital place area (OPA), parahippocampal place area (PPA), and retrosplenial complex/medial place area (RSC/MPA). All are characterized by higher responses to scene images compared to other stimuli, and they seem to contain information relevant to navigating through the 3D world [5–7].

Here, we focus on assessing whether these areas contain information about surfaces and 3D spatial layout per se rather than other visual properties of scenes. One important challenge is differentiating low- to mid-level visual features from the 3D information that they ultimately support. Scene-selective cortical areas could be responsive to configurations of 3D surfaces per se, but these tend to co-occur in the natural world with cues such as patterns of orientations and spatial frequencies.

Foundational work on scene areas has employed hypothesis-driven stimulus conditions such as preserving the same surfaces in a scene while selectively disrupting their configuration [2]. These manipulations cannot (and don't pretend to) completely remove the influence of orientation and spatial frequency, since these basic features are the building blocks of all more complex images. Instead, they focus on finding a theoretically interesting level of tolerance to low-level feature variation: after finding higher PPA responses even for unfurnished rooms compared to isolated objects from those rooms, they sought to test whether the presence of a coherent configuration of surfaces was the critical aspect of the stimulus driving responses. They created two conditions: (1) fractured images (surfaces cut apart and separated by white space) vs. (2) fractured + re-arranged surfaces: the same fractured surfaces but re-arranged within the image to prevent them being perceived as a 3D space. Fractured surfaces, which still had a recognizable 3D structure, drove PPA responses similarly to the empty scenes – at least in terms of overall univariate magnitude averaged over the entire region. In contrast, univariate responses to the fractured + re-arranged surfaces were significantly lower than both real scenes and fractured scenes. While low-level features were not perfectly teased apart from the features of interest – e.g., the fractured rooms still contained edges oriented the same way in similar parts of the image – these features have a close relationship to surface information in everyday life. Thus, this comparison was taken as a powerful demonstration that PPA cares about configurations of surfaces, insofar as intact 3D structure – even without the walls being connected – led to greater univariate activity than a scrambled version that broke 3D structure. Notably, however, this study and similar early work [8] did not attempt to quantitatively compare how much of the variance in voxel responses could be attributed to the 3D structure of depicted scenes vs. other competing models.

More recent studies have investigated what aspects of the 3D structure of scenes vs. competing models are contained in more fine-grained patterns across cortex. These studies compared spatial information of interest with patterns of orientation and spatial frequency information captured by a GIST baseline model [9]. For example, Henriksson, Mur, & Kriegeskorte [10] found evidence for an OPA representation of 3D surfaces of a scene beyond GIST features. Bonner & Epstein [6] found that OPA in particular captures information about navigational affordances in scenes. A notable aspect of both of the above findings was the surprising amount of information corresponding to GIST features (even more than the affordance features in [6]), perhaps reflecting the close relationship observed between GIST features and spatial and/or category information that may be important for scene-selective areas [9,11,12]. Given this close relationship, considering GIST features to be a "low-level" baseline may result in some meaningful information about the 3D shape of a scene being discounted—a possibility that complicates interpretations by "counting against" models of interest, specifically when GIST features perform well.

This in-principle issue might matter less in practice, however, if a model of the 3D structure of a scene performed decisively better than a set of high-performing baseline models. Lescroart & Gallant [13] constructed 3D models that captured spatial characteristics of major surfaces (distance and orientation in 3D space) in videos of moving scenes (see Fig 1 for stimulus example). The best-performing 3D model was compared against the best-performing of three baseline models, each capturing static information about orientation and spatial frequency that was computed from stills of the videos. The high performance of the 3D vs. baseline models appeared to resolve the theoretical issues described above: while the GIST model had performed close to or better than the models of interest in the previous studies [6,10], Lescroart & Gallant's 3D model performed decisively better than even a Gabor wavelet baseline model, which itself performed more than twice as well as the GIST baseline model. Gabor wavelet models have previously been successful at predicting neurophysiological and fMRI data from early visual cortex (EVC), and they have also served as baseline models for higher-level visual cortical representations of static image stimuli [14]. However, the combination of *moving* stimuli with a *static* Gabor baseline model also raises concerns: while the video stimuli contain motion parallax as a depth cue, this baseline model did not model low-level motion cues correlated with motion parallax (e.g., horizontal motion in the lower plane of the screen that covaries with surfaces closer in space). This leaves open the possibility that the dominance of the 3D model that was observed in OPA/PPA/RSC does not demonstrate evidence of 3D information *per se*, but instead may

**A** *Henriksson, Mur, & Kriegeskorte (2019)*
*Static scene images, 3 textures x 32 spatial layouts*

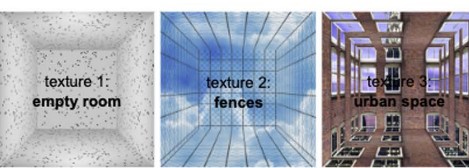

**Baseline model: GIST**
*OPA:* 3D surface model info > GIST info (GIST info surprisingly high)

**B** *Bonner & Epstein (2017)*
*Static scene photograph stimuli*

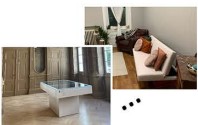

**Baseline model: GIST**
*OPA:* GIST model info > local scene affordance info

**C** *Lescroart & Gallant (2019)*
*Moving video stimuli, 3D rendered*

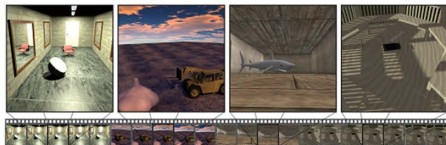

**Baseline model: static gabor wavelet model (1,425 features)**
*OPA, PPA, RSC:* 3D surface model info > static Gabor wavelet info

**D** *Present experiment:*
*Static scene photograph stimuli*

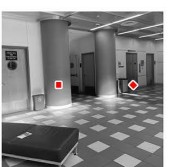

**Baseline model: static gabor wavelet model (300 features)**
*OPA, PPA, RSC:* static Gabor wavelet info > 3D surface model info

**Fig 1. Stimuli and baseline models used in 3 related studies (A-C) as well as in the present experiment. (A)** Henriksson, Mur, & Kriegeskorte (2019) used 3D-rendered static scene image stimuli and a GIST baseline model. In OPA, they found greater 3D surface model info, along with surprisingly high GIST information. **(B)** Bonner & Epstein (2017) found evidence for local scene affordance information, which may be related to the 3D-surface information discussed here. They used static scene photograph stimuli, a GIST (static) baseline model, and in OPA found greater GIST than local scene affordance information as well as some information uniquely attributable to local scene affordances. (Actual stimulus examples have been replaced with similar images of indoor scenes for copyright reasons.) **(C)** Lescroart & Gallant (2019) used 3D-rendered moving video stimuli, used a static gabor wavelet baseline model, and in OPA, PPA, and RSC found 3D information beyond their baseline model. **(D)** The present experiment uses static scene photograph stimuli, a static Gabor baseline model, and found in OPA, PPA, and RSC that a Gabor-wavelet baseline model accounts for more variance than a 3D surface-model. (Note that instead of an actual stimulus image, we have used a similar one that we own the copyright for).

reflect variance that is attributable to un-modeled low-level motion information that co-varies with 3D scene-surface features. Note that this possibility would not by itself argue against the presence of 3D information in these regions—instead, it would further highlight the importance of building a more nuanced picture of covariation between "3D" and "2D" feature sets (for a more detailed explanation, see *Discussion: Implications for the visual system*).

In the present work, we characterized the performance of 3D scene-surface models [13] and their relationship to the orientation and spatial-frequency features they covary with in natural images. First, we wanted to ensure that our baseline model encompassed the full set of low-level features we wished to discount. We could have done this by either A) using an expanded baseline model to account for the low-level motion features present in video stimuli or B) using static photograph stimuli instead of video stimuli so that it was unnecessary to include low-level motion information in our baseline model. We chose to use static image stimuli (Fig 1D) with a static gabor-wavelet model (capturing orientation and spatial-frequency information), in part to maintain more contact with prior work that used static orientation and spatial-frequency features as a baseline [e.g., 9,11,12]. In addition, using photographs allowed us to characterize the performance of 3D scene-surface models using natural images, as opposed to the computer-rendered stimuli used in prior

work. While this choice sacrifices one aspect of "naturalness" of the stimulus set (motion), the photographs appear more "natural" in another sense in that they are more lifelike than stills from a computer-rendered video. We then compared the static Gabor-wavelet baseline model with two 3D surface models in OPA/PPA/RSC, each capturing the distance to and 3D directions of surfaces in each image: (1) a model capturing summaries of these 3D characteristics across the entire image (as in Lescroart & Gallant, 13; "3D-global model"), and (2) a model summarizing these 3D features within each quadrant of the image, preserving more fine-grained spatial information ("3D-quadrant-based model"; Fig 2A). Importantly, we used an algorithmic stimulus selection procedure based on ground-truth 3D scene properties to select a set of images that could maximally differentiate between the 3D-global- and the 3D-quadrant-based-models. We verified that this procedure also minimized the relationship between Gabor-wavelet baseline features and each 3D model (Fig 2B).

Thus, our modeling approach, coupled with the use of static images, allowed us to investigate the generalizability of previous 3D scene information in OPA/PPA/RSC in two ways. First, because our stimuli consisted of static images, our baseline Gabor wavelet pyramid model included the key types of low-level features to be discounted. Second, because the stimuli were photographs rather than computer-rendered, the image statistics and the environments depicted in each image were consistent with what participants might encounter in the real world.

## Results

In this study, we compared the performance of two 3D-surface models vs. a Gabor-wavelet baseline model in predicting human voxel responses to photographs of real-world scenes. We recorded fMRI data while participants viewed images from the Taskonomy training set (Fig 2C–2E). On each trial, participants were asked to make one of two judgments: which of two dots placed on the image indicated the part of the scene closer to (/farther from) the viewer, or which of the dots was a square (/diamond). Note that because we do not focus on potential task-related differences in the present manuscript, we present the main results collapsed across tasks. The task alternated across runs (Fig 2E, Methods). To quantify the 3D vs. Gabor-wavelet-baseline information in each voxel, we fit each of three encoding models (Fig 2A) with cross-validated ridge regression (Fig 3A) and performed variance-partitioning analyses (Fig 3B and 3C) to determine how well each model uniquely predicted voxel responses. Finally, we evaluated the robustness of our results across several different pre-processing and voxel-selection decisions.

Fig 3A shows the cross-validated prediction performance of each voxelwise encoding model in each visual region. As expected based on prior work in other labs, we observed that in early visual areas, both a Gabor-wavelet baseline model [17] and a model capturing distances to and orientations of surfaces in 3D space ("3D-global model") predicted voxel-wise activity above chance [10,13] (all $p$s < 0.0003, uncorrected, two-tailed; see S2A Table for stats corresponding to individual tests). The magnitude of the raw cross-validated predictions was similar to Lescroart & Gallant's [13].

For scene-responsive regions OPA/PPA/RSC, we found that while prediction performance was significantly above chance for all three models (all $p$s < 0.014; Fig 3A and S2A Table), the Gabor-wavelet-baseline model unexpectedly outperformed the 3D models (Fig 3A; all ps < 0.0026) in each scene region, demonstrating significant effects in the opposite direction as in ref. [13]. (Note that while neither our nor ref. [13]'s conclusions depend on across-ROI comparisons, we include a supplementary across-ROI ANOVA (S3 Table) to reassure readers that power in our analysis approach is adequate to detect such differences.) The greater Gabor-wavelet-baseline performance prompted us to apply a 3D-vs.-Gabor-wavelet-baseline variance-partitioning analysis in each region (Fig 3B and 3C) to determine the unique contributions of each set of features [13]. We adopted this approach because the small amount of 3D information we found could either be due to 3D features per se, or it could be due to imperfect separability of 3D vs. Gabor-wavelet-baseline features in our stimulus set, which would mean that information that is genuinely Gabor-wavelet related could contribute to a small degree to the fit of the 3D model. Surprisingly, we only found evidence that the *Gabor-wavelet baseline* features uniquely predicted responses in these regions (unique Gabor-wavelet-baseline info in variance partitioning vs. 3D-quadrant-based features: OPA: p = 0.0006; PPA: p = 0.0002; RSC: p = 0.026; unique

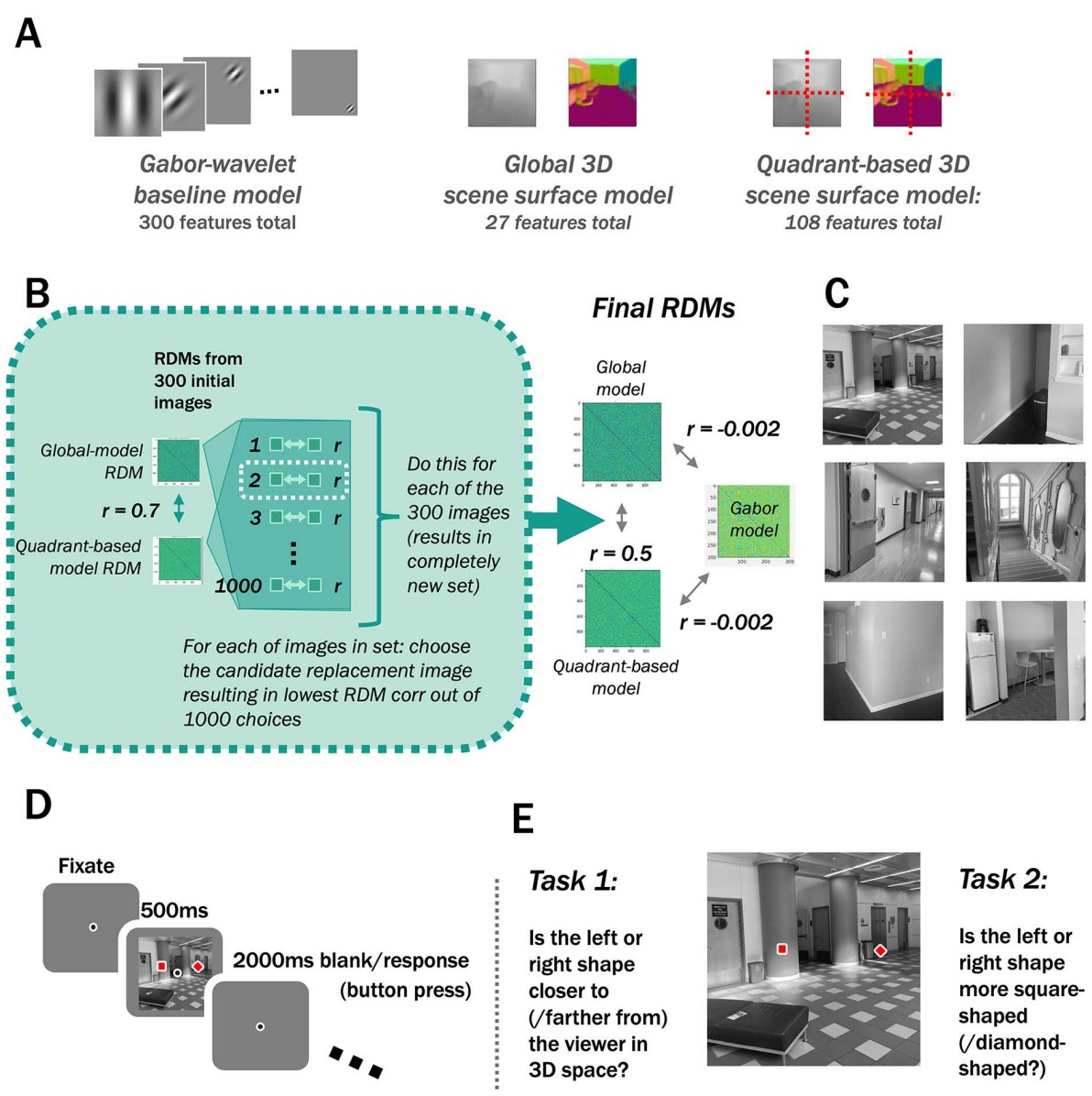

**Fig 2. Methods. (A)** Gabor-wavelet baseline and 3D feature sets were computed similarly to Lescroart & Gallant (2019), with 3D features based on the Euclidean-distance and surface-normal annotations included in the Taskonomy [15] training set. **(B)** The 300 stimulus images were chosen to minimize 3D-global- vs. 3D-quadrant-based-model RDMs. First, an initial set of stimulus images was chosen iteratively (similarly to Groen et al., [16]), from a pool of images with maximally different depth information in the image's 4 quadrants. The resulting initial image set had an RDM correlation of 0.7 between the quadrant-based and global 3D models. Next, we replaced each of the 300 stimulus images with the image from 1000 random choices that resulted in the lowest RDM correlation. This stimulus selection procedure incidentally yielded a low correlation between the Gabor-wavelet baseline model and each of the two 3D models. While our main analyses used voxelwise encoding models, we used RDMs for our stimulus selection to quantify feature variation across images. **(C)** Examples of images chosen to be similar to stimulus images. Candidate stimuli were drawn from the Taskonomy [15] training set, consisting of one image from each of 679,000 unique camera locations. **(D)** Example trial. **(E)** Tasks. On alternating runs, participants either answered 1) whether the left or right shape was closer to (farther from) the viewer in 3D space or 2) whether the left or right shape was more square-shaped (diamond-shaped). Note that in the main analysis we collapse across these two task types, but the results are similar when each condition is considered individually.

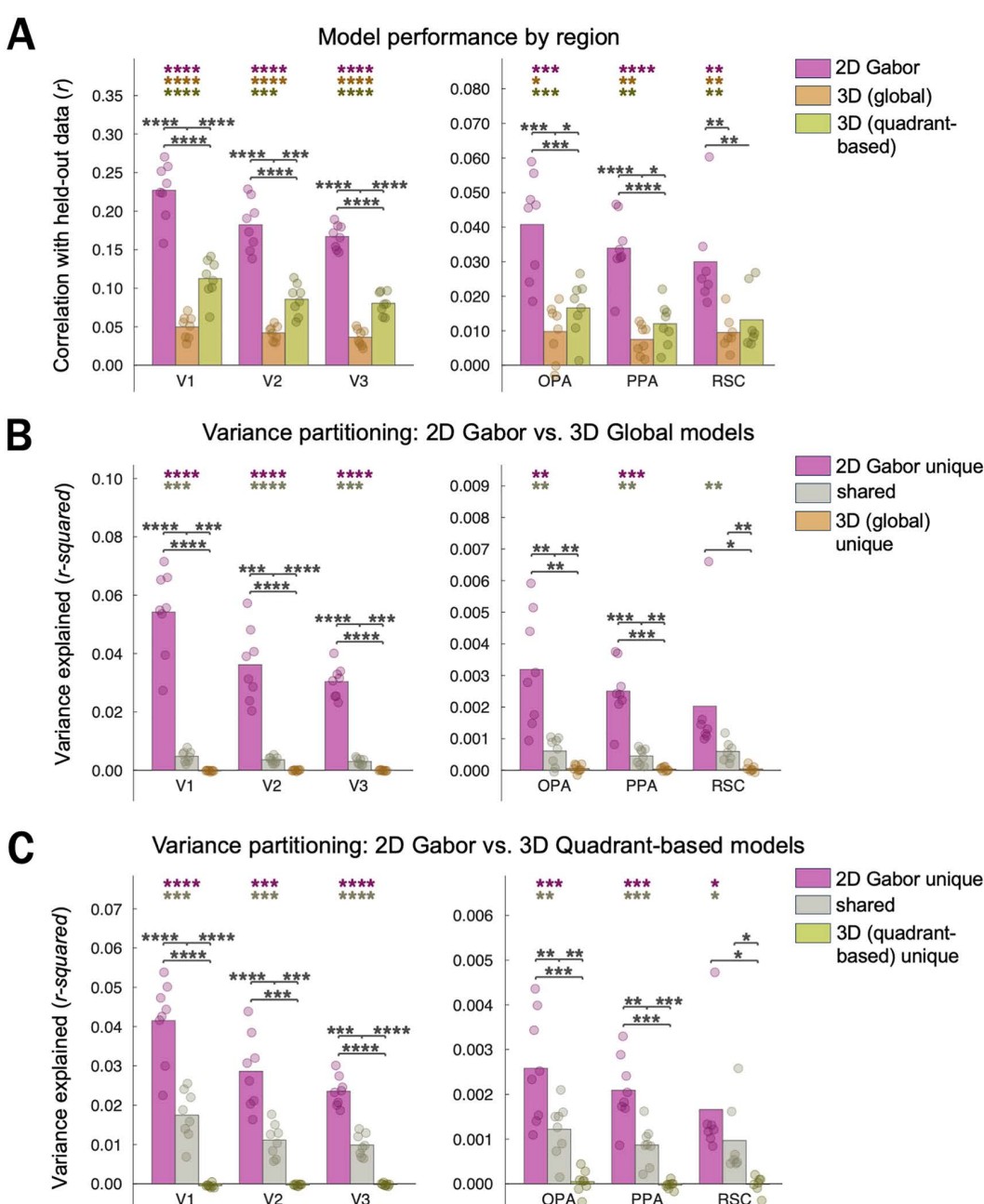

**Fig 3. Raw model performance (correlation of model predictions vs. held-out data) and variance partitioning of voxelwise encoding models.** Dots represent individual participant data. Stars denote significant two-tailed effects in the positive direction (****=p<0.0001; ***=p<0.001; **=p<0.01; *=p<0.05, uncorrected). Colored stars correspond to individual tests, and brackets indicate tests between pairs of conditions. For main analyses, we report raw correlations and variance explained, without normalizing to a noise ceiling. Note that magnitudes across ROIs are not interpretable, but patterns within ROIs are. **(A)** Prediction performance by model. **(B)** Variance partitioning results for "2D" Gabor-wavelet-baseline model vs. 3D-quadrant-based model. **(C)** Variance partitioning results for "2D" Gabor-wavelet-baseline model vs. 3D-global model.

Gabor-wavelet-baseline info in variance partitioning vs. 3D-global features: OPA: p = 0.0018; PPA: p = 0.0005; RSC: p = 0.053). In this analysis, shared variance represents information not uniquely attributable to either model. In each area, there was significant *shared* 3D-quadrant-based/Gabor-wavelet-baseline (OPA: p = 0.0019; PPA: p = 0.0004; RSC: p = 0.026) and *shared* 3D-global/Gabor-wavelet-baseline (OPA: 0.0069; PPA: 0.0018; RSC: 0.0048) representations, with no significant evidence for uniquely 3D representations (3D-global model: all $p$s > 0.12; 3D-quadrant-based model: all $p$s > 0.41; Fig 3B and 3C and S2B and S2C Table).

Note that RSC results follow the same pattern as the OPA/PPA results (significant unique-Gabor and shared variance, without significant unique-3D variance) but are not as robust, consistent with previous work. In OPA and PPA, the unique variance explained by the Gabor-wavelet-baseline model was significantly greater than either the 3D-quadrant-based model (ps < 0.0008) or the 3D-global model (ps < 0.004). In addition, unique Gabor-wavelet-baseline model variance was greater than the shared variance not uniquely attributable to either model: unique Gabor > shared variance between Gabor-wavelet-baseline and the 3D-quadrant-based model (ps < 0.0027) and unique Gabor-wavelet baseline > shared variance between the 2D gabor-wavelet model and the 3D-global model (ps < 0.0061). Again, RSC showed a similar pattern of greater Gabor-wavelet-baseline performance but was less reliable overall (Fig 3B and 3C and S2B and S2C Table).

Next, we investigated whether these results were robust to methodological decisions such as voxel selection. First, it could have been the case that by including up to 200 voxels in each category-selective ROI (possibly a less stringent cutoff than other studies, e.g., [13]), we might have ended up including voxels with more selectivity to lower-level vs. category-selective features. If this were the case, it might account for the higher responses we found to the Gabor-wavelet baseline model than to the 3D scene-surface features. In Fig 4A–4C, we plot model performance and variance partitioning results as a function of selecting different numbers of voxels in each ROI (the N voxels with the highest scene > object functional-localizer activation). This plot demonstrates a strikingly consistent pattern of relative model performance across voxel counts, arguing that our results are not an artifact of our voxel selection decisions (see *Methods: Regions of Interest*). We also note that, in hindsight, our decision to include up to 200 voxels in each ROI may have lowered the *overall magnitude* of performance for all models, especially in scene regions. However, the baseline Gabor-wavelet model is also strikingly dominant for the voxel counts that maximize overall model performance.

A second follow-up analysis is related to work demonstrating that scene-selective regions occupy regions of cortex that overlap with retinotopically defined visual regions [18,19]. Since the relationships between response properties of these overlapping regions have not yet been fully characterized, it is in theory possible that the dominance of the Gabor-wavelet baseline model could have been due to voxels from meaningfully distinct but overlapping brain regions influencing our results in scene-selective cortex. While our main analyses (Fig 3) excluded any voxels from scene regions that were also included in V1-V3, we performed an additional analysis that also excluded any voxels that were identified as part of regions potentially overlapping with scene regions: V3AB, IPS0, hV4, or LO. This version of the analysis, including individual-subject data points, is almost indistinguishable from our main results (S2 Fig).

We also tested whether the pattern of higher Gabor-baseline vs. 3D information could have been partly due to not rescaling model performances by voxel noise ceilings, a choice made due to details of our experimental design (see *Methods: Ridge regression*). The most straightforward consequence of this choice is that the raw values would be lower than if they were rescaled by noise ceilings, especially for regions with less reliable responses. Because this first consequence would only be expected to change each ROI's overall magnitudes and not the relative performance of models within ROIs, it would not affect our main conclusions. A second consequence of analyzing raw prediction performances is that each voxel's raw correlation or variance contributed equally to the ROI average, irrespective of its noise ceiling. If voxels with higher 3D model performances had also had systematically lower noise ceilings, using the raw values could have weighted these voxels less than in reference [13] and contributed to our pattern of higher Gabor-baseline performance.

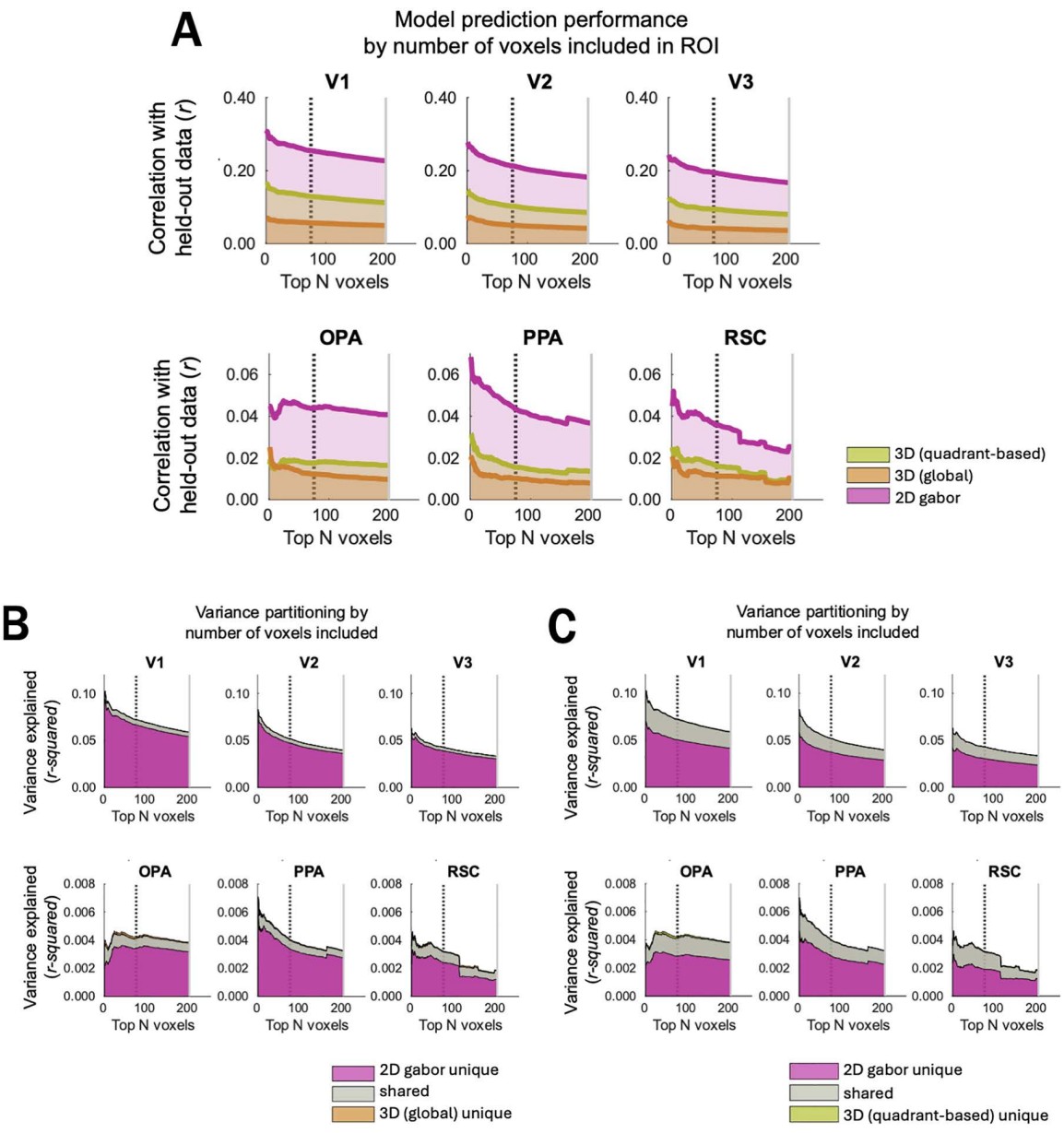

**Fig 4. Raw prediction performance and variance partitioning results plotted as a function of number of voxels included in each ROI.** Voxels were ordered by independent-localizer activation: scrambled > baseline for EVC ROIs, top rows; scenes > objects for scene-selective ROIs, bottom row. Then, results were plotted as though we'd defined a 1-voxel ROI, a 2-voxel ROIs, etc. The downward slopes of many plots indicate that our ROI sizes may have caused us to find a smaller magnitude of effects than in other studies that used a stricter cutoff. ROIs could include up to 200 voxels (light gray vertical line), with a minimum of 75 voxels (dark dashed vertical line) required for an ROI to be included in the analysis. **(A)** Prediction performance of each model by number of voxels included. The prediction performance corresponds to the height of the line on the y axis. **(B)** Variance partitioning for the 3D-global vs. 2D-Gabor-wavelet baseline models. For variance partitioning plots, the height of each *shaded region* corresponds to the variance attributable to each condition. The "shared" condition corresponds to variance not uniquely attributable to either model. **(C)** Variance partitioning for the 3D-quadrant-based vs. 2D-Gabor-wavelet baseline models. The height of each *shaded region* corresponds to the variance attributable to each condition.

However, individual-participant surface maps (Figs 5, 6, S3 and S4) show that very few voxels in visual cortex had higher 3D vs. Gabor-baseline model performance, so this does not seem likely. Indeed, when we rescaled by noise ceilings, the main findings were almost completely unchanged (Fig 7A–7F). One exception was that one ROI (PPA) showed significant unique 3D information for one of the two 3D models (global model; Fig 7D, PPA), although we note that the magnitude was only slightly different from the original version, and the pattern of greater unique Gabor information was even more reliable than in the original analysis (Fig 7D vs. 7C). Together, these analyses and visualizations show that our pattern of dominant Gabor-wavelet model performance does not seem to be an artifact of voxel-selection details or rescaling by the noise ceiling for each voxel.

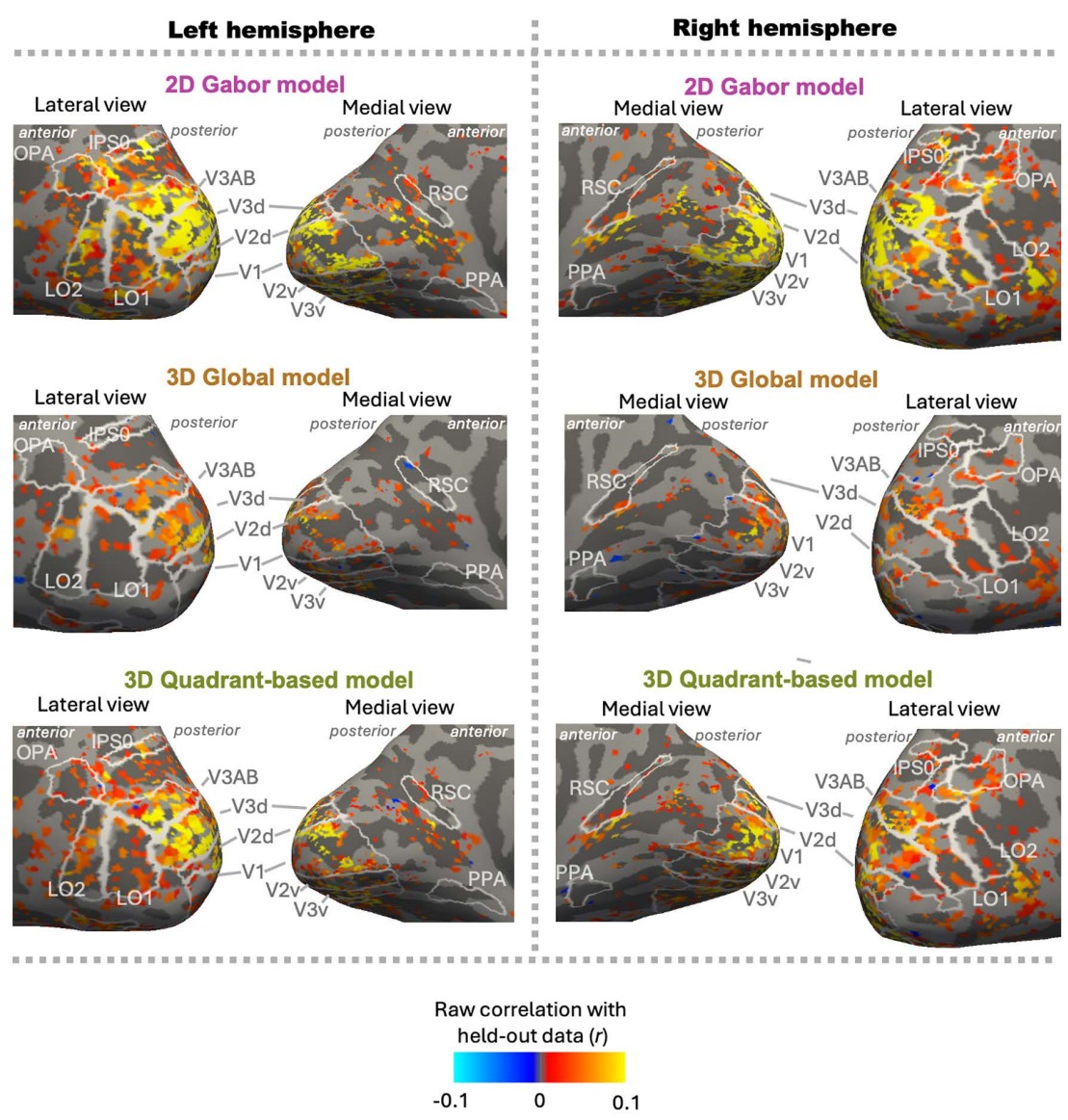

**Fig 5. Model performance results for an example participant (S01) across the cortical surface.** (See S3 Fig for other subjects.) Positive correlations are denoted by warm colors. Negative correlations are denoted by cool colors. White outlines indicate the boundaries of each region of interest. Voxels with significant noise ceilings (p<0.05) are included, after which model-performance p-values are FDR-corrected (q<0.05 shown here).

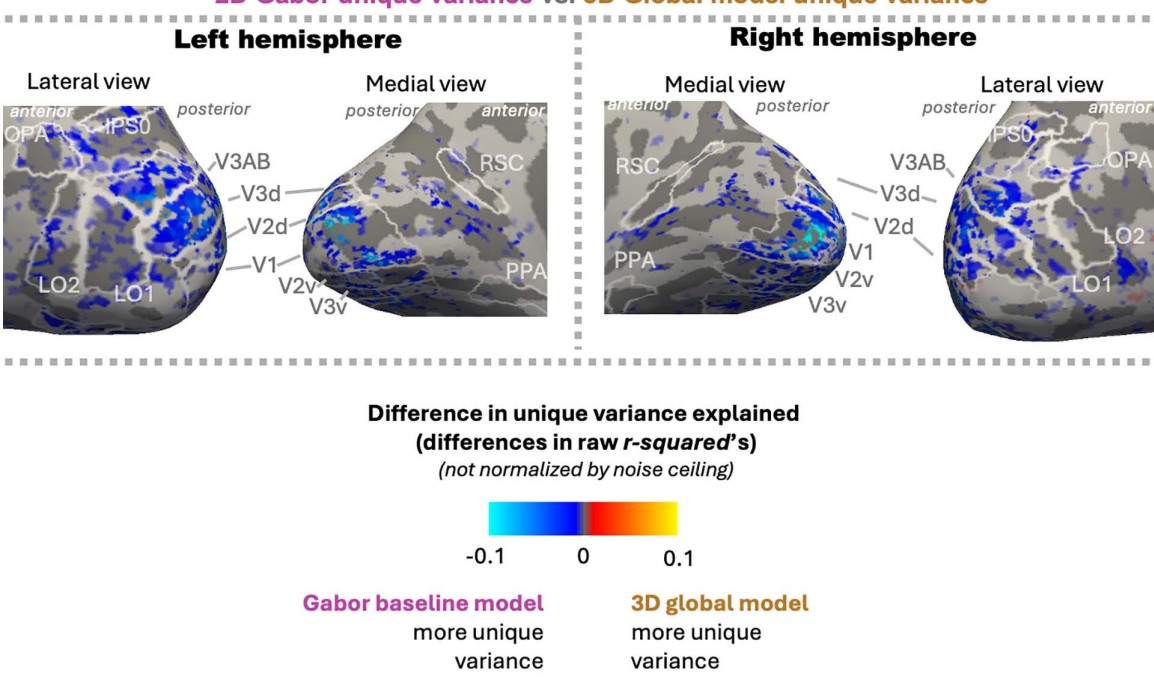

**Fig 6. Variance-partitioning results for an example participant (S01) across the cortical surface (see S4 Fig for other subjects).** Cool colors denote more unique variance explained by the Gabor-wavelet baseline model, and warm colors denote more unique variance explained by the 3D-global model. White outlines indicate the boundaries of each region of interest. Voxels with reliable noise ceilings (p < 0.05) are included, after which model-performance p-values are FDR-corrected (q < 0.05 shown here).

Next, it is possible that the timecourse of prediction performance for the Gabor-wavelet baseline model peaked earlier than for the scene-surface model, and that we had by chance chosen too early of a time window for our analyses to detect 3D scene-surface information. Fig 8A–8C shows a sliding window analysis in which model performance is compared across different ranges of TRs. All timepoints show the same pattern of dominant performance by the Gabor-wavelet model.

Finally, while we collapse across two tasks in our main analyses (one for which the 3D structure is relevant and one for which it is not), each task alone shows an almost identical pattern of model performances in all ROIs (Fig 9A–9F).

Together, these results show a pattern of higher Gabor-wavelet-baseline performance compared to 3D scene-surface models in scene-responsive cortical regions OPA, PPA, and RSC. This pattern of results was robust to changes in several methodological decisions, participants' task, and the variation of the 3D model that we chose. Furthermore, none of our five sets of follow-up analyses showed statistically or even numerically greater performance for either scene-surface model compared to the Gabor-wavelet baseline model in any ROI.

## Discussion

The present work investigated whether 3D scene-surface information in OPA, PPA, and RSC (e.g., Lescroart & Gallant, [13]) is robust to comparisons with a baseline model that includes the relevant types of low-level information (in this case, static orientation and spatial frequency info) present in the stimuli (in this case static scene photographs). We included two variations of a 3D scene-surface model: *3D-global* (summarizing across the entire image) and *3D-quadrant-based* (summarizing across quadrants of the image), comparing each against a *Gabor-wavelet baseline model* capturing orientation and spatial-frequency information. We found higher prediction performance for the Gabor-wavelet baseline model

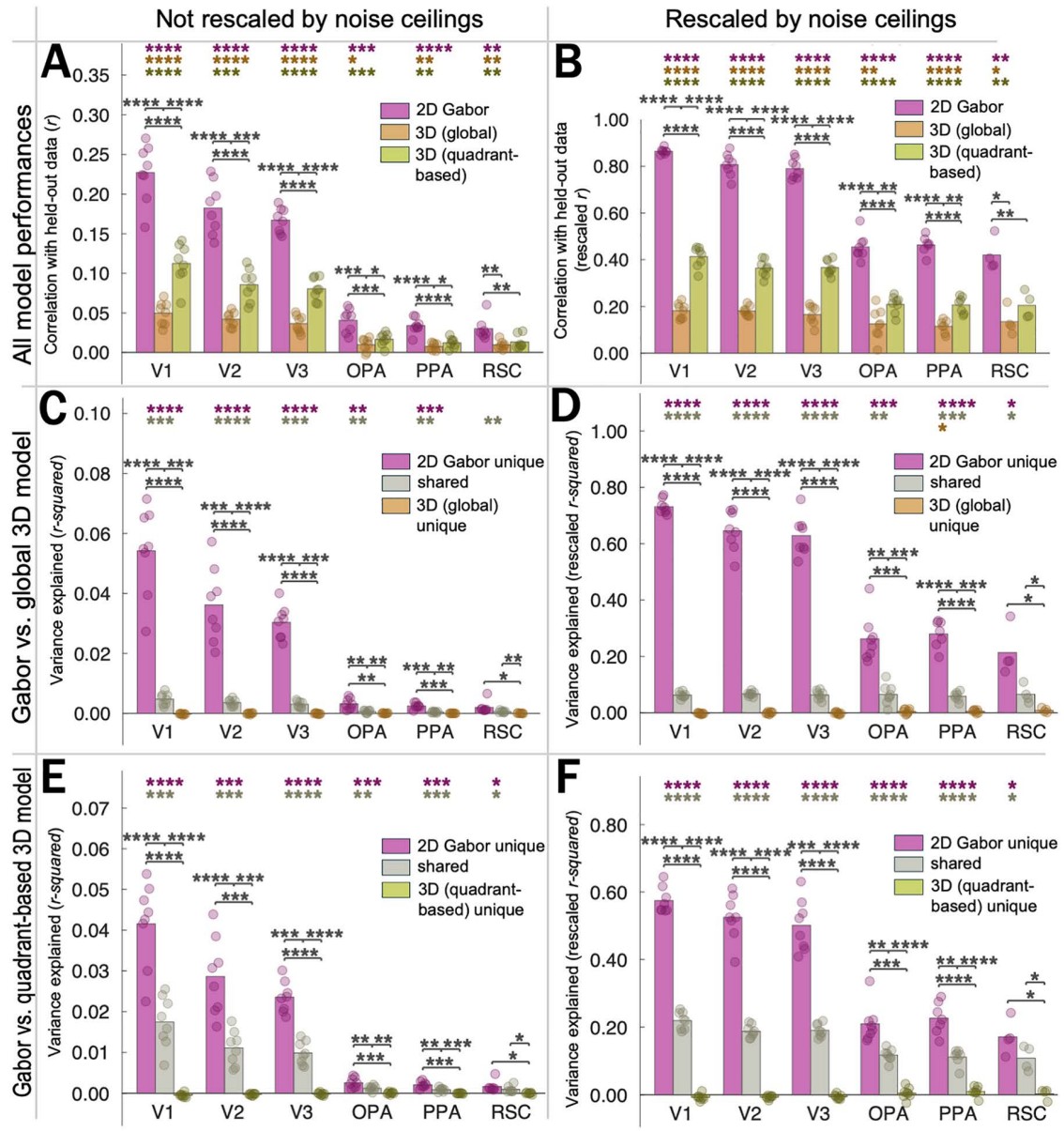

**Fig 7. A comparison of raw (A,C,E) and noise-ceiling-rescaled (B,D,F) prediction performances (A-B) and variance-partitioning results (C-F).** While the relative *magnitudes* of results in scene regions (OPA, PPA, RSC) are increased compared to those in early visual cortex (V1-V3), the pattern of within-ROI model comparisons and statistical tests is largely unchanged. We note that when rescaling by noise ceilings, 3D-unique information crossed the threshold to significance in one scene ROI (PPA), for one model (3D Global; cf panels C, D). However, we also note that the statistical tests comparing the 3D model's smaller magnitude vs. those of the 2D-unique and the shared information also became more significant. Note also that the data for panels with A, C, and E has been re-plotted from Fig 3 for easier comparison with noise-ceiling-corrected data in panels B, D, and F.

compared to each of the 3D-scene-surface models in scene-responsive cortex. In each of these areas, we also found evidence for unique Gabor-wavelet-baseline representations (representations uniquely attributable to this model in a variance partitioning analysis) but not unique 3D representations, as well as significant differences between the two. While these results appear contrary to the dominant claim about OPA in particular [5–7,13], as well as findings across all three regions [13], we discuss potential reasons for this discrepancy below.

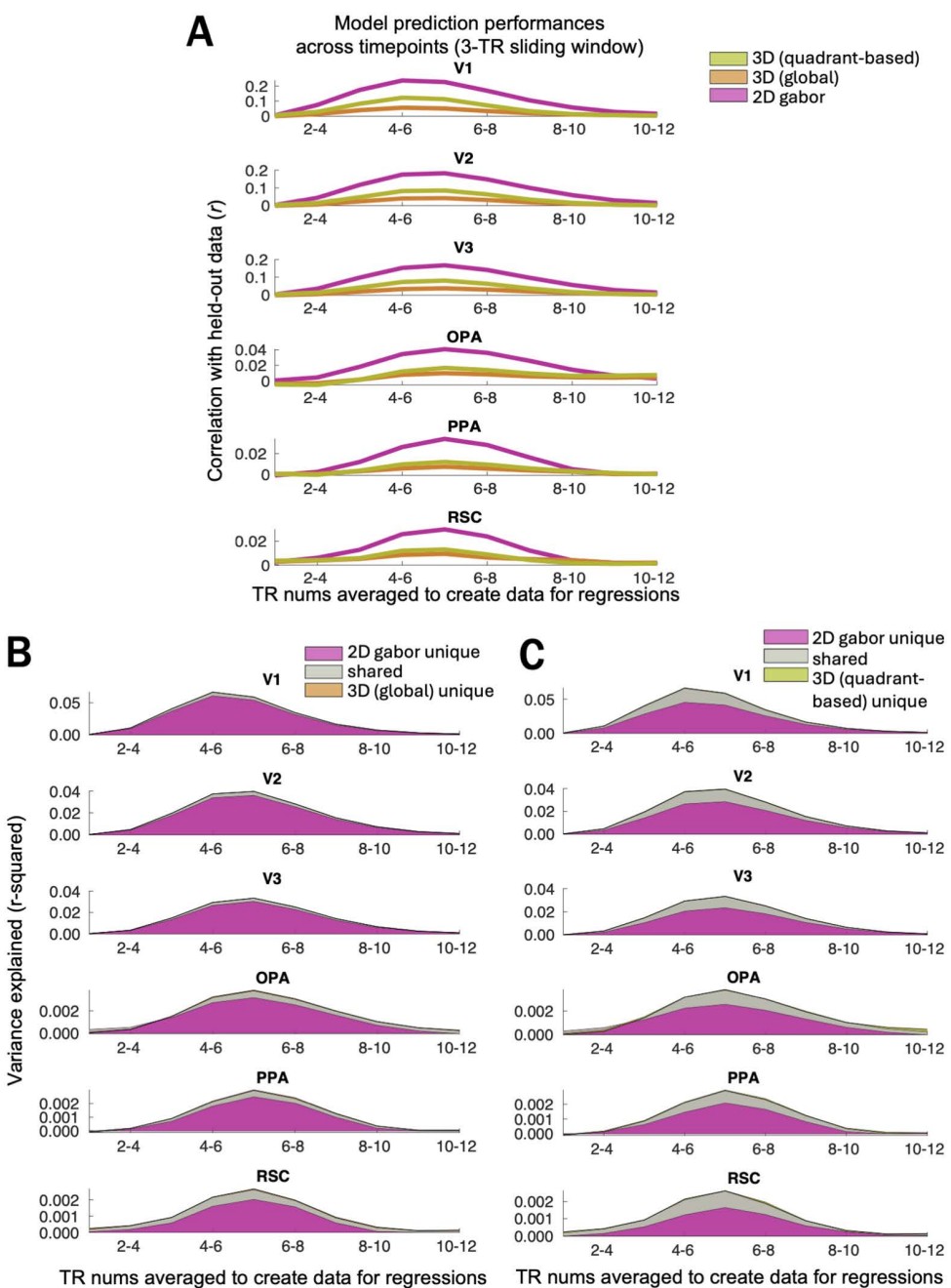

**Fig 8. Raw prediction performances (A) and variance partitioning results (B-C) calculated across ten 3-TR windows.**

## More information uniquely attributable to a Gabor-wavelet baseline model than to 3D scene-surface models

In each scene region, our main variance partitioning analysis revealed significantly greater information uniquely attributable to the Gabor-wavelet baseline model than to either scene-surface model. This was the case in almost all of our follow-up analyses as well, with the exception of the task-breakdown analyses, where for several comparisons within RSC, Gabor model performance was numerically but not significantly higher than 3D model performance. We also found

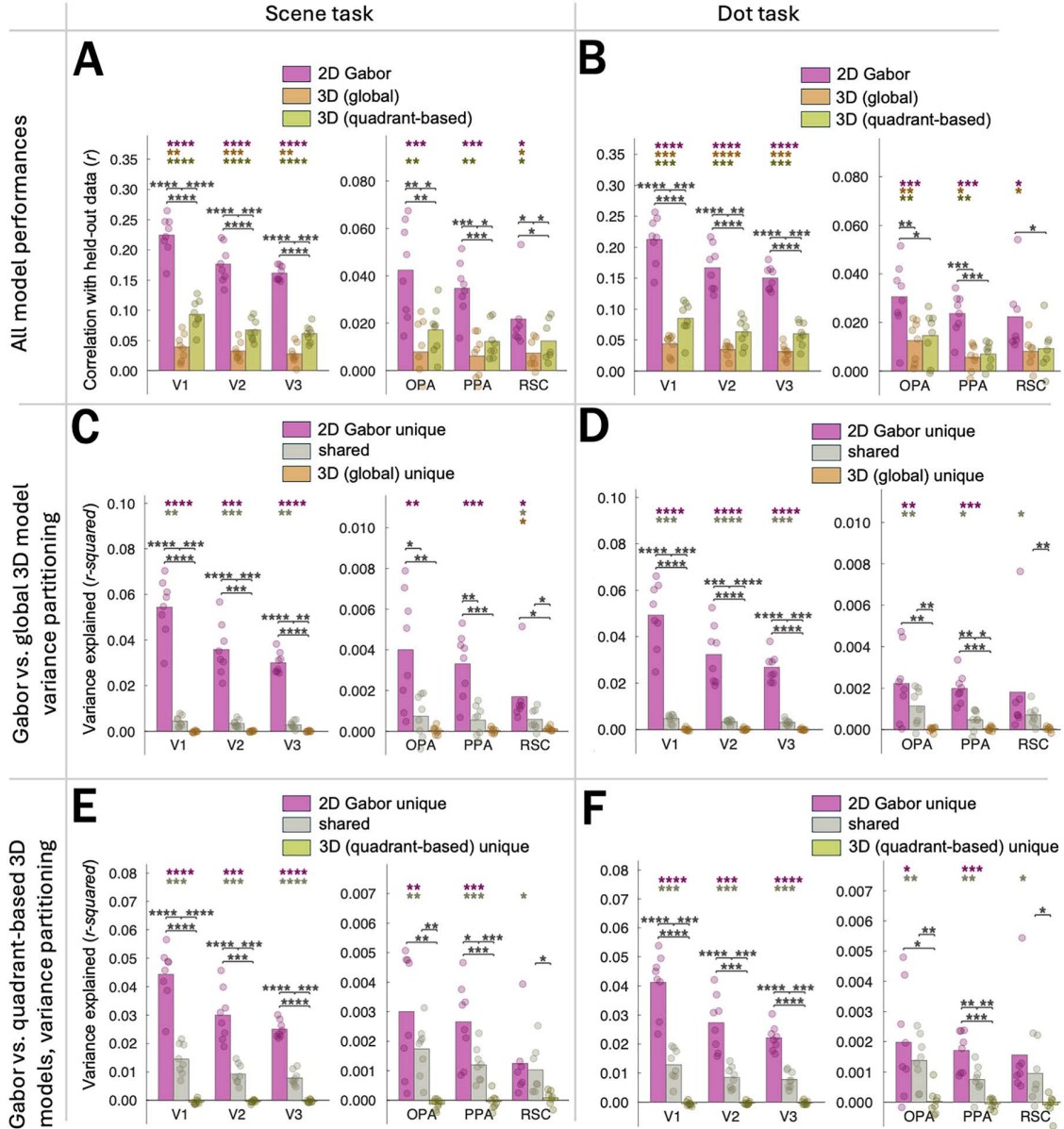

**Fig 9. Raw prediction performances (A-B) and variance partitioning results (C-F) split according to task.** "Scene task" (A,C,E): participants responded which dot was closer to the viewer in 3D space. "Dot task" (B,D,F): participants responded which dot was a square/diamond. Tasks varied across separate runs, and identical pairs of dots appeared in both task conditions.

little to no evidence that 3D scene surface features uniquely explained any activity in scene-selective cortex, despite consistent Gabor-baseline information. These results stand in contrast with the conclusions reached by Lescroart & Gallant [13], who examined responses to moving videos of scenes. As suggested in the introduction, the higher performance of the 3D-surface model within each scene region in [13] could be partially attributable to contributions from low-level motion cues that could not be accommodated by the static Gabor-wavelet model used as a baseline. This is because, just like the static Gabor information Lescroart & Gallant [13] discounted, motion information can also correlate with 3D surface information present in stimulus videos. Our use of static photographs removed motion information from the stimulus set,

allowing us to use a static Gabor-wavelet baseline model that accounted for the low-level information in the images (orientation and spatial frequency). With this change, we observed that the Gabor-wavelet baseline model generally outperformed the 3D model within each scene region, demonstrating significant results in the opposite direction as in ref. [13]. In sum, our use of static images sacrificed the more dynamic aspects of human visual experience in order to simplify our choice of baseline model. Future work could use video stimuli with a baseline model that also captured low-level motion information. This complementary set of findings would lead to a richer understanding of how more diverse types of 3D information and corresponding lower-level features are represented in scene-responsive cortex.

In addition to including motion information, the stimuli used by Lescroart & Gallant [13] also employed lighting and texture differences to disrupt the relationship between 3D surfaces and low-level visual features like orientation and spatial frequency. The importance of this issue is illustrated by imagining that rightward-facing surfaces usually co-occur with down-right oriented lines in one part of the image and/or up-right orientations in a different part of the image (etc. for other surface orientations). The more these feature sets co-occur across images, the more difficult it would be for regression models to disentangle the variance explained by each. By rendering their scenes using a 3D modeling software, Lescroart & Gallant [13] introduced high-contrast shadows and other visual features to disrupt these co-occurrences within individual scenes. We took a different approach: algorithmically selecting a set of naturalistic scene photographs from a large image database. While each individual photograph still contained lifelike visual features, the *set* of photographs was selected to reduce covariation between feature sets, capitalizing on naturally occurring disruptions in these co-variations. Two aspects of our data argue that our differing results were not due to a failure to reduce co-variation between feature sets. First, we find lower shared variance between these models (variance not uniquely attributable to either model) compared to Lescroart & Gallant [13]. Second, even if we imagined all shared variance being attributed uniquely to the 3D surface model, the Gabor-wavelet baseline model would still explain more variance than the 3D surface model. (It is possible to visualize this by seeing that for OPA/PPA/RSC in Fig 3C and 3D, the heights of the "unique Gabor" bars are higher than the height that would be seen by stacking the corresponding "3D unique" and "shared" bars).

We also investigated the robustness of our results to several different methodological decisions. First, when varying ROI size, we still see greater Gabor-wavelet performance than 3D-model performance (Fig 4). While our main results investigate voxel responses within a narrower time window than Lescroart & Gallant's [13] and without the added flexibility of FIR features, our pattern of results is consistent across a wide range of timepoints (Fig 8A–8C). Finally, we see the same pattern for each of our two tasks (Fig 9A–9C) as we do using data collapsed across tasks (main results).

It is also worth noting that both the present study and Lescroart & Gallant [13] fit models using cross-validation, making both sets of results robust to overfitting. In addition, while one of our 3D models was smaller than Lescroart & Gallant's due to the selection of fewer distance bins (9 vs. 3; resulting in 27 and 108 features in our 3D models vs. 91 features in Lescroart & Gallant's), it seems unlikely that this change was a major factor in the reversal of results. This is first because one 3D model in [13], without any distance information at all, performed nearly as well as the full 3D model [13; Fig 3A; model had 10 features, vs. 27 and 108 in our 3D models]. Even with this reassurance, we still reduced the number of features in our Gabor-wavelet-baseline model (300 vs. 1425 in [13]) to mitigate any unanticipated effects of relative numbers of model features. Another difference between studies is that our stimuli consisted primarily of indoor scenes. We considered this an important test of 3D-surface models, which are designed to capture 3D configurations of surfaces like walls and floors that are especially common in indoor scenes. While this stimulus set allowed us to use ground-truth distance measurements to compute 3D-surface features, it also meant that neither of our 3D models contained a sky channel due to inherent limits in depth sensors (see Methods and [15]). While we cannot completely rule out a critical role of the sky channel in driving the success of Lescroart & Gallant's full 3D model, it is notable that a model without sky pixels was the second-best-performing of their three 3D models [13, Fig 3A], and the first two PC's of voxel responses in the full model's feature space do not appear to trivially capture the presence of sky pixels [13, Fig 6]. Thus,

we don't believe this is a likely reason for the difference in results. Finally, different stimulus sets may also contain different distributions of global properties (e.g., openness [12]), which could differentially covary with each model. This possibility motivates future work investigating the impact of more stimulus sets, performance metrics, and model details on these findings.

While our conclusions may at first seem at odds with the field's present understanding of scene-responsive cortex, two prior studies also used a baseline model capturing spatial frequency and orientation information (a GIST model [9]) and found results compatible with ours. First, Bonner & Epstein [6] investigated local scene affordances, or representations of the parts of a visual scene that a viewer could traverse within the depicted 3D space. They found higher performance for GIST features [6] than for local-affordance (arguably 3D) information. Note that while this work did not explicitly test the presence of 3D surface information, local-affordance information is related to 3D surface information since configurations of surfaces within a scene constrain where humans are able to walk. While they also found a small amount of unique local-affordance information, the presence of this information is also compatible with our results, especially considering Lescroart et al.'s [13] data showing a Gabor wavelet model performing over twice as well in OPA/PPA/RSC as a GIST baseline model. Similarly, Henriksson et al. [10] also used a GIST model as their baseline for quantifying 3D scene surface information. A higher-performing Gabor-wavelet baseline model might be expected to out-perform the 3D information in this data set as well. In light of this converging pattern of results, it is important to note that the consistently high (and sometimes higher) performance of "2D" vs. "3D" features doesn't necessarily argue that these regions don't represent the "3D" features in a meaningful way (see "*Implications for the visual system*", below).

Our results are also consistent with results reporting variations in the sizes of voxel receptive fields (vRFs) and spatial-frequency selectivity within and across regions. First, scene regions tend to have larger voxel receptive fields (vRF's) [e.g., 20] than regions of early visual cortex [e.g., 21], and our Gabor-baseline model includes smaller receptive fields than both the Global and the Quadrant-based 3D models. While these vRF properties do not make predictions about 2D vs. 3D feature differences per se, they do predict a smaller advantage for the Gabor vs. 3D models in scene regions vs. EVC regions when considering the regions of visual space captured by the features of each model. In a follow-up ANOVA, we found a significant ROI x Model interaction supporting this across-ROI prediction (Fig 3 and S3 Table). Several previous studies have also found selectivity profiles for spatial frequencies in scene regions that are consistent with these regions being well described by a Gabor model [22–24]. Other work has found that while radial frequency (capturing curvature information) appears stronger in early- to mid-level regions vs. higher-level regions, spatial-frequency information similar to that captured by our Gabor-baseline model is relatively preserved across all regions [25]. Our observation that scene-selective regions are well described by Gabor model is thus compatible with this finding and adds to the substantial literature on spatial-frequency selectivity in scene-selective cortex.

In sum, when performing model comparisons between higher-level models (like 3D surfaces and affordances) and "low-level" models (like GIST or Gabor wavelet models), it is critical that baseline models include all relevant types of low-level information (e.g., motion energy when using videos)—especially low-level information that covaries with the higher-level features. Omitting low-level feature types from the baseline model undermines one of the largest advantages of model comparison: the ability to interpret quantitative differences in the variance attributable to each model. And, as noted above, even though it is simpler to include all relevant low-level feature types when using static image stimuli, future work could leverage the advantages of richer stimuli by accounting for features such as motion cues in videos.

### Implications for the visual system: Do our results argue that "low-level" or "2D" information is dominant in scene-selective cortex?

The present pattern of results, in combination with prior work, highlights the dominance of orientation and spatial-frequency information—often considered 2D, or low-level—in scene-selective cortex. Within the variance-partitioning framework used here and in previous work, it might be tempting to argue that scene areas aren't really *for* representing

3D information in scenes. Importantly, however, patterns of orientation and spatial frequency can actually serve a critical role in 3D scene understanding. For example, these features are computationally sufficient to infer global scene properties related to 3D layout [26], and human behavior does indeed seem to rely on them for higher-level scene processing tasks [11]. Thus, while it is hard to argue that a *single* oriented Gabor patch could support 3D processing, *combinations* of orientations and spatial frequencies across the visual field provide a richer set of cues that could add up to meaningfully 3D information. Of course, in the extreme case, matching the spatial distribution of all "low-level" orientation and spatial-frequency features of a grayscale image would result in a perfect copy of the image, which would of course also preserve all mid- and higher-level properties. Considering a wider range of visual features (e.g., including color, motion, or binocular disparity) would not escape this close relationship since the new "low-level" features would then also need to encompass the entire set of features available to be combined into cues to these higher-level properties.

Thus, in the present work, the Gabor-wavelet baseline model could have performed well in scene-responsive cortex *by virtue* of capturing meaningful 3D information, in the form of specific combinations of 2D orientations and spatial frequencies that serve as cues to the 3D shape of the space. More broadly, this highlights the point that *individual* model features with interpretable verbal labels, like the filters in a Gabor-wavelet model, do not necessarily add up to a feature set that is similarly interpretable *as a whole*. This adds a layer of complexity to any attempts to compare the relative amounts of "higher-level" information vs. any corresponding "lower-level" information that could serve as cues.

## How can we meaningfully characterize the format of higher-level visual information?

Model comparison remains a critical tool for precisely quantifying the degree to which different types of information are present across the brain. The present work demonstrates that the full benefits of this approach depend on better understanding the information captured by each model and how model features covary — a point that has been widely acknowledged when comparing feature sets derived from artificial neural networks [27]. Critically, we argue for this focus to be broadened even to models like Gabor-wavelet pyramids, whose *individual* features can be easily described with verbal labels.

One potential path forward is to focus on finding the most theoretically meaningful cutoff between "higher-level" 3D spatial features and the corresponding "lower-level" 2D features used as a baseline condition. This has been a successful approach in studying visual responses to real-world objects compared to their corresponding mid-level features [28]. However, an analogous approach for 3D scene understanding may be more challenging. First, object categories can be reported in one word, making it relatively quick to confirm that features in the baseline condition are not sufficient for object recognition. This is not the case for 3D scene layout, which would likely slow the iterative process of finding appropriate feature sets and stimuli. Second, since 3D scene understanding is comparatively less studied than object recognition, there is a risk of over-investing in understanding specific feature sets before establishing a broader picture of their relationships.

More recently, DNNs have made it easier to generate a variety of visual feature sets without relying on a researcher to explicitly parameterize them. This could be especially important for scene-related spatial features, which may be more difficult to verbalize compared to object features. Two recent studies have made headway in this direction. Dwivedi et al. [29] and Wang et al. [30] tested how well human brain responses could be predicted by the internal representations of several DNNs [15] that were trained to predict 3D properties of scene images (e.g., surface normals) vs. 2D or semantic properties of images (e.g., 2D edges or scene category, respectively). They each found promising relationships with brain representations for features from 3D vs. 2D DNNs, setting the stage for future work to more directly interpret the content of these feature sets. This is an important contribution to the endeavor of finding the most theoretically meaningful cutoff between features that are meaningfully interpreted as "3D" and those that are best understood as 2D baseline features.

A second potential approach, either before or instead of finding a binary cutoff between "lower-level" and "higher-level" feature sets, would be to develop a better understanding of how features covary in naturalistic visual input. To the extent

that certain feature sets reliably covary, even features that seem to be "lower-level" could be used by the visual system as 3D or other "higher-level" information. For example, if certain patterns of orientation and spatial frequency reliably cue a down-right-oriented ceiling, they could support effective 3D behavior. Note that the strength of this approach depends both on the size of an image set and how well its image statistics reflect naturalistic visual experience. In a very small image set, covariation between feature sets might be driven by the specific stimuli that happened to be picked by chance and might not reflect the cues that are available to us in real life. Even larger stimulus sets are not guaranteed to capture naturalistic feature covariations. For example, image sets structured around semantic categories of visual objects or scenes may emphasize the cues that are most important for object or scene *recognition* rather than spatial understanding. Even choosing more broadly from images people post on the internet could introduce specific biases, like image symmetry, that could disrupt feature covariations. A stronger emphasis on understanding the statistics of naturalistic visual inputs could ultimately lead to high-throughput testing of a large set of models without relying on custom stimuli for specific feature sets. It could also help avoid assumptions about which visual regions or feature sets could support certain naturalistic behaviors—for example, it could be the case that even representations in early visual areas might contain reliable and behaviorally meaningful 3D structure information.

Overall, we believe that using both these broader and more targeted approaches together will allow model-comparison approaches to be a more effective tool for understanding different types of visual representations across cortex.

## Conclusion

The present work examines the strength of 2D and 3D representations in scene-responsive regions of cortex (PPA/OPA/RSC) while participants viewed naturalistic scene-photograph stimuli. In contrast to some previous work, we find that a Gabor-wavelet baseline ("2D") model compares favorably to explicit models of 3D information in scene-responsive cortex, even when accounting for potentially shared variance. This work highlights the difficulty in drawing conclusions about separability of 3D (or higher-level) representations from the patterns of orientation and spatial frequency that underlie them. This motivates future investigations of exactly which information is "3D" or meaningfully related to higher-level properties of scenes.

## Methods

### Ethics statement

**Participants.** 8 participants (6 female, 2 male) between the ages of 25 and 31 participated in the experiment. The protocol was approved by the Institutional Review Board of The University of California, San Diego, and all participants gave informed written consent. Each participant completed a behavioral training session lasting approximately 1.5 hours, followed by a retinotopic mapping session and three fMRI sessions where they performed the main task. Each scanning session lasted approximately 2 hours. Participants were compensated $15/hr for the behavioral session and $25/hr for each of the three scanner sessions.

**fMRI data acquisition and pre-processing.** fMRI data collection was completed at the UC San Diego Keck Center for Functional Magnetic Resonance Imaging, on a General Electric (GE) Discovery MR750 3T scanner. Functional echoplanar imaging (EPI) data were collected using a Nova 32-channel head coil (NMSC075-32-3GE-MR750) and the Stanford Simultaneous Multi-Slice (SMS) EPI sequence (MUX EPI), with a multiband factor of 8. This resulted in 9 axial slices per band (72 slices total; 2mm$^3$ isotropic voxels; 0mm gap; matrix = 104x104; FOV = 20.8 cm; TR = 800ms; TE = 35ms; flip angle = 52°; in-plane acceleration = 1).

We acquired a high-res T1 anatomical scan in the same session (GE ASSET on a FSPGR T1-weighted sequence; 1x1x1 mm$^3$ voxel size; 8136ms TR; 3172ms TE; 8° flip angle; 172 slices; 1mm slice gap; 256x192cm matrix size), corrected for inhomogeneities in signal intensity using GE's Phased array uniformity enhancement (PURE) method. This was used for segmentation, flattening, and delineation of retinotopic visual areas.

Preprocessing was completed using Freesurfer and FSL (available at https://surfer.nmr.mgh.harvard.edu and http://www.fmrib.ox.ac.uk/fsl). We used Freesurfer's recon-all utility [31] to perform cortical surface gray-/white-matter segmentation of each subject's anatomical scan. These segmentations were then used to define cortical surfaces on which we delineated retinotopic ROIs used for subsequent analyses in EVC (see *Regions of Interest*). We used Freesurfer's manual and automatic boundary-based registration tools [32] to generate transformation matrices that were then used by FSL FLIRT [33,34] to co-register the first volume of each functional run into the same space as the anatomical image. Motion correction was performed using FSL MCFLIRT [34], without spatial smoothing, with a final sinc interpolation stage, and with 12 degrees of freedom. Finally, slow drifts in the data were removed using a high-pass filter (1/40 Hz cutoff). No additional spatial smoothing was performed for main task runs. After pre-processing, we z-scored each voxel's timeseries within each run and epoched the data based on the start time of each trial. Because of the short ITI and fast event-related design, we averaged responses between 2.4 and 4.8 seconds after stimulus onset to use as each trial's data for our main analyses, unless specified otherwise.

**Procedure and experimental conditions.** For all task runs described here, stimuli were projected onto a semi-circular screen 21.3 cm wide and 16 cm high, fixed to the inside of the scanner bore just above the participant's chest. The screen was viewed through a mirror attached to the head coil, from a viewing distance of 49 cm. After taking into account the shape of the screen and the square stimuli, this resulted in a vertical extent of approximately 18.1° (max vertical eccentricity of 18.1°/2). The background was a mid-gray color, with a darker gray placeholder outline marking the location of the square stimuli between stimulus presentations. The fixation point was a black rounded square of 0.2° with a white outline and was on the screen throughout each run.

We collected a total of 12 experimental runs (514 TRs each; 6 minutes 52 seconds long) in each of the three sessions, for a total of 36 runs per participant. Runs alternated between two tasks (see *Tasks* section below), with task order counterbalanced across participants. Within each run, three mini-blocks corresponded to narrow, medium, or broad regions of the scene ("spatial spread") that target shapes could appear in (see *Dot Selection* below). The order of mini-blocks within each run was counterbalanced within participants to minimize order effects of spatial-spread conditions. Four combinations of response mappings were counterbalanced across participants along with task order, for a total of one combination per participant.

We used a fast event-related design, with each stimulus presented for 500ms on each trial, with a 2000-ms un-jittered response window/ITI.

**Stimuli and tasks.** *Main task:* This experiment was originally designed to test the robustness of different types of 3D information to changes in task and spatial attention. On each trial, a pair of red shapes (one square and one diamond) was superimposed on a grayscale scene photograph. In *distance judgment* runs, participants were instructed to respond whether the left or the right shape was on the part of the scene that would be closer to (/farther from) the viewer in three-dimensional depth if viewed in real life [35,36]. In *shape judgment* runs, the participant was instructed to respond whether the left or right shape was a square (/diamond). Participants responded "left" (square/diamond/closer/farther) using their index finger and "right" using their middle finger. Each run was organized into three mini-blocks designed to manipulate the spread of participants' spatial attention. In each mini-block, dots could either be constrained to a narrow, medium, or broad portion of the stimulus image. Order of the mini-blocks was counterbalanced within participants. (See *Dot Selection*, below, for more details of dot selection.)

All responses were made with their right (dominant) hand. Response mappings were consistent throughout the entire session for each participant, including during the 90-minute behavioral practice session. To ensure that any task differences were not due to different difficulties of tasks, the diamond and square could be parametrically adjusted to be more rounded, allowing us to continuously adjust the difficulty of the task to match the distance judgment task. Dot pairs were selected to keep participants off ceiling in the distance judgment runs so that difficulty could be accurately matched. (See S1 Fig for participant performances across task.) By choosing two tasks that depended on judgments of two dots, we

aimed to avoid task effects that could be trivially explained by small eye movements or covert spatial attention. While not analyzed here, each image could appear with one of three different pairs of dot locations with varying spatial extent. (See "Procedure and experimental conditions" for details.)

*Functional localizer task:* During the localizer task, participants viewed blocked presentations of images of scenes, objects, faces, bodies, scrambled images (images divided into small square pieces and randomly re-arranged), as well as a fixation baseline condition. They pressed a button when they detected trials in which the same image was identical to the image immediately previous (one-back task). Stimulus presentation code was adapted from code from the Grill-Spector Lab.

**Stimulus selection.** The 300 stimulus images were selected from the Taskonomy training set [15] based on an original goal of this study: to maximally differentiate between the 3D-global and 3D-quadrant-based scene-surface models. The Taskonomy training set is made up of 4.5 million scene images from hundreds of unique buildings, with a range of different camera locations in each building. We first subsetted this data set by selecting one image from almost all unique camera locations, resulting in 679,000 images in our starting set. We began with an iterative approach similar to Groen et al.'s [16], creating representational similarity matrices (RDMs) based on the cosine distance between features of each image, starting from a pool of images with maximally different depth information in each of the image's 4 quadrants. Iterating through randomly chosen stimulus *sets* resulted in a plateaued correlation of ~0.7 between 3D-global and 3D-quadrant-based feature RDMs. We next iterated through each of the 300 images in this initial set, testing RDM correlations for 1000 potential replacement images and using the replacement that resulted in the lowest RDM correlation of the set. After replacing each of the 300 images once, we were left with a correlation of 0.5 between 3D-global and 3D-quadrant-based feature RDMs. Although this procedure did not explicitly orthogonalize either 3D feature set against gabor features, we incidentally ended up with smaller RDM correlations (3D-global vs. gabor: -0.002) than Lescroart et al.'s [13] validation data set (0.274). This pattern was robust to the choice of distance metric (see S1 Table). While our main analyses used voxelwise encoding models, we used RDMs for our stimulus selection to quantify feature variation across images.

**Dot selection.** For each stimulus image, we chose locations of 3 pairs of superimposed dots (the square- and diamond-shaped dots described in the *Tasks* section above): one each within a narrow, medium, and broad spread of potential dot locations. We did this using a semi-automated Matlab script, indicating the allowed region of the image for each spatial spread condition and prompting the selector to (if possible) select via mouse click one potential set in which the correct answer was "left closer" and one in which the correct answer was "right closer". Feedback was given in the command window after each selection to help the selector choose dot locations with roughly similar average differences in distance across dot pairs in the narrow, medium, and broad spatial spread conditions. To choose the final set from these manually selected options, a Matlab script chose from these options to contain an equal number of left-correct and right-correct dot pairs and verified that the narrow, medium, and broad conditions didn't differ substantially in the depth difference between pairs of dots.

**Feature sets: Gabor wavelet baseline model.** The *Gabor-wavelet model* contained 300 features. Each feature had one of 5 spatial frequencies (0, 2, 4, 8, and 16 cycles per image), and non-zero spatial frequencies had one of 4 orientations (0, 45, 90, or 135 degrees). The stimulus image was tiled by a grid of gabor wavelets for each spatial frequency. Note that there is a dependence between the spatial frequency and tile size: higher spatial frequencies are used for smaller tile sizes. We also compared fitting success in one pilot subject using a 1,425-feature gabor model, achieving similar results as with the 300-feature model. Thus, we opted to use the smaller model for all main analyses. These features were generated using code from https://github.com/gallantlab/motion_energy_matlab.

**Feature sets: 3D scene surface models.** 3D-g*lobal* and 3D-*quadrant-based scene surface features* were computed using the corresponding ground-truth distance and surface-normal (surface direction) image maps included with our stimulus images, which were selected from the training set for the Taskonomy [15] neural networks. These distance

and surface-normal image maps were generated from images taken with RGB-depth sensors. Information from multiple camera rotations and viewpoints of the same indoor space was used to improve the accuracy of these individual measurements. To generate features from these maps, we adapted code from Lescroart & Gallant [13], first grouping pixels in the distance maps into 3 bins, spaced so that distance bins were roughly equally represented across images. Using the surface-normal maps, we grouped each pixel into one of 9 surface-direction bins: forward, upward, downward, leftward, rightward, up-left, up-right, down-left, and down-right. In the *3D-global scene-surface model*, we counted the proportion of pixels in each image falling into each combination of the 3 distance and 9 surface-direction bins for a total of 27 features. In the *3D-quadrant-based scene-surface model,* we next divided the image into 4 quadrants, with each of the 108 resulting features corresponding to a combination of image quadrant x surface direction x distance bin. Before model fitting, features were z-scored.

**Ridge regression.** We fit models using ridge regression, with nine cross-validation folds. For the analyses collapsing across task, each of the 9 left-out sets was made up of 4 of the 36 runs (two from each task). For the scene-task-only results, each of the left-out sets was made up of 2 of the 18 same-task runs. To choose ridge parameters, we nested 10 inner cross-validation folds within the training data of each main fold and chose the lambda predicting the highest r-squared among the inner cross-validation folds. This lambda was then used to fit all of the data in the training portion of the main fold. There were 13 possible ridge parameters: 0, as well as 12 values logarithmically spaced between $10^{-2}$ and $10^{5}$. Early analyses on pilot data showed that this ridge parameter selection matched performance using a more extensive set of ridge parameter choices.

Our main analyses report prediction performance, first averaged across the 9 main cross-validation folds, then averaged across all voxels in an ROI. We did not normalize by a noise ceiling for our main analyses. This means that magnitudes of effects *across* ROIs may vary due to factors like SNR across ROIs. Statistical significance was computed via permutation tests, shuffling the test data 10,000 times relative to the predicted data before completing the same process of averaging prediction performance across cross-validation folds and across voxels in each ROI. This generated permuted values corresponding to each ROI and participant, which we then used to calculate a null distribution of t-statistics across participants. We tested individual model performances against zero by calculating a t-statistic from the actual data and comparing that to the null distribution. We report two-tailed uncorrected p-values without assuming a symmetrical null distribution.

For the follow-up analysis investigating the effects of noise-ceiling rescaling, noise ceilings were calculated according to the method in Allen et al. [37], since we had fewer exact stimulus repeats than the ~five recommended for the method used in [13]. Significance of the noise ceilings was calculated via a permutation test with 2000 permutations.

For follow-up visualizations showing results on inflated surfaces, we first stored model-fitting and variance-partitioning results in native volume space and performed FDR-correction ($q < 0.05$) across all voxels with a significant noise ceiling ($p < 0.05$). (Note that we did not rescale these results by noise ceiling.) We then transformed each set of results from each subject's native volumetric space to their native inflated-surface space. We used Freeview's command-line tools and GUI [38] to visualize these results.

**Regions of interest.** For regions V1, V2, and V3, we followed previously established retinotopic mapping protocols [39–41]. Initial masks for areas V1, V2, and V3 were manually drawn based on retinotopic mapping data collected in a separate session, and candidate voxels were selected for further analyses based on a scrambled > baseline contrast (see Stimuli & Tasks: Functional Localizer Task, above), using a false discovery cutoff of $q < 0.05$. We used a combination of localizer runs from past experiments in our lab and the present experiment (stimuli from Grill-Spector lab), for a total of 4–10 functional localizer runs per subject. For OPA, PPA, and MPA/RSC, initial masks were manually drawn around contiguous clusters of voxels in each subject's native space, including voxels with $q < 0.05$ for a scenes > objects contrast. ROI definitions were mutually exclusive, and any voxels included in V1-V3 were excluded from scene ROIs. For a supplementary analysis (S2 Fig), voxels included in any of 5 additional regions—V3AB, hV4, IPS0, LO1, and LO2—

were additionally excluded from scene ROIs. To ensure that any differences in performance across ROIs were not due to averaging over dramatically different numbers of voxels, we capped our main analyses to the 200 voxels with the strongest localizer signal in each bilateral ROI; we included ROIs with>= 75 voxels bilaterally, resulting in 7/8 participants with all ROIs defined and one with all defined except MPA/RSC (see Fig 4 for main analysis results as a function of number of voxels included). This participant is omitted from the MPA/RSC analyses.

For follow-up analyses investigating the role of noise-ceiling rescaling, voxels passing the functional-localizer threshold were further required to have a significant noise ceiling value ($p < 0.05$) to be included in the final ROI. As in the main analyses, a given ROI was required to contain 75 voxels bilaterally to be included in this analysis. Because there was not perfect overlap between voxels meeting the noise-ceiling threshold and the functional-localizer threshold, fewer voxels passed this combined cutoff in some ROIs. This resulted in 8/8 participants with usable EVC and OPA ROIs, 7/8 participants with a usable PPA, and 4/8 with a usable RSC.

**Deviations from original analysis plan.** This study was designed to test whether the advantage for the 3D model vs. the Gabor-baseline model generalized from ref. [13] to a new stimulus set and, if so, whether the 3D model's advantage was sensitive to task manipulations (see *Stimuli and Tasks: Main Task* for more information on these manipulations) or to details of the 3D model. Because we did not find higher 3D vs. Gabor-baseline performance overall (Fig 3) or in either individual task condition (Fig 9A), we do not focus on across-task comparisons in this work. Instead, after our initial analyses, we collapsed across tasks to maximize power for all further Gabor vs. 3D model comparisons.

We have also added follow-up analyses in response to reviewer suggestions: 1) creating individual-subject flat maps for variance-partitioning and model-performance results; 2) generating variance-partitioning or model-performance results for analysis versions in which we had previously only generated one or the other; 3) plotting model performance for the dot task separately (instead of just separating out the scene task); 4) versions of analyses that excluded voxels from scene regions if they were included in any of five additional retinotopic regions, as in [13] (see Figs 5 and S2); and 5) versions of the analysis in which each voxel's model performances are rescaled by its noise ceiling, as in [13] (see Fig 7). To improve legibility, we also decided during the second round of review to modify the original plots so that they showed scene regions' performance on a separate axis from EVC regions' performance, as well as to auto-plot significance markers for single conditions and across-condition comparisons in all plots. We felt that these plotting updates were important for demonstrating the reliability of the raw correlation and variance-partitioning data in scene regions, despite their overall lower *magnitudes* compared to EVC. (These overall lower magnitudes are likely due largely to SNR, combined with the decision not to rescale by noise ceilings for our main analyses; see Fig 7 to compare with noise-ceiling-rescaled results).

## Supporting information

**S1 Fig. Behavioral performance.** Participants' performance was well above chance, with no obvious outliers. The difficulty of the scene task was fixed because it depended on pre-determined placement of dots on the scene. We chose these placements with the goal of keeping participants off of ceiling. We were able to staircase performance in the dot task to match the scene task, avoiding large differences in performance across runs and enabling us to determine that participants were remaining alert.
(PDF)

**S2 Fig. Main model-performance and variance-partitioning results, additionally excluding V3AB, IPS0, LO1, LO2, and hV4 from scene-selective ROIs OPA, PPA, and RSC.** Results are almost identical to main results, including individual data points (cf. Fig 3, for which only V1-V3 voxels are excluded from scene-selective ROIs).
(PDF)

**S3 Fig. Surface maps of individual participants' model performance results for all three models.** Main analyses were done in volume space, and surface-space maps are for visualization purposes only. However, we have FDR-corrected significance values across all voxels with significant noise-ceiling results. (Note, however, that these analyses, like the main analyses, do not rescale model performaces by noise ceilings.) Color bar depicts the difference between the unique variance explained by the 3D global model (positive, warm colors) and the Gabor-wavelet baseline model (negative, cool colors). Panels A-G include surface maps for each of the 7 subjects not included in the main-text Fig 5 (S02-S08).
(PDF)

**S4 Fig. Surface maps of individual participants' variance partitioning results for the 3D global model vs. the Gabor baseline model.** Main analyses were done in volume space, and surface-space maps are for visualization purposes only. Color bar depicts the difference between the unique variance explained by the 3D global model (positive, warm colors) and the Gabor-wavelet baseline model (negative, cool colors). Panels A-G include surface maps for each of the 7 subjects not included in the main-text Fig 5 (S02-S08).
(PDF)

**S1 Table. RDM correlations between 2D gabor-wavelet model and 3D-global scene-surface models are reliable within stimulus sets across 3 distance metrics.** For the present stimuli, only the 3D-global scene-surface model was included, as it most closely corresponded to Lescroart & Gallant's (2019) model [13].
(PDF)

**S2 Table. Statistical tests for main analyses in text.** A) Two-tailed p-values corresponding to permutation tests for each model compared to 0, as well as comparisons of performance for each. B) Variance partitioning, two-tailed p-values corresponding to permutation tests for 3Dquadrant-based and Gabor models. C) Variance partitioning, two-tailed p-values corresponding to permutation tests for 3D-global and Gabor models.
(PDF)

**S3 Table. A repeated-measures ANOVA on model performances across ROIs.** ANOVA contains main effects of ROI and Condition (Model), as well as the interaction between them. Not only do we find extremely reliable main effects of ROI and model, but critically we also find an extremely significant interaction between the two. While the Gabor model is the strongest model in every region, this interaction reflects the magnitude of the *Gabor advantage* decreasing from early visual regions to scene regions (Fig 3A).
(PDF)

## Acknowledgments

We thank Mark Lescroart for sharing data and code from his 2019 paper, as well as helpful discussion and feedback on pilot analyses. We thank members of the Serences Lab for help with data collection and useful discussions.

## Author contributions

**Conceptualization:** Anna Shafer-Skelton, Timothy F Brady, John T Serences.

**Data curation:** Anna Shafer-Skelton.

**Formal analysis:** Anna Shafer-Skelton.

**Funding acquisition:** Anna Shafer-Skelton, Timothy F Brady, John T Serences.

**Investigation:** Anna Shafer-Skelton.

**Methodology:** Anna Shafer-Skelton, Timothy F Brady, John T Serences.

**Project administration:** Anna Shafer-Skelton, John T Serences.

**Resources:** Anna Shafer-Skelton, Timothy F Brady, John T Serences.

**Software:** Anna Shafer-Skelton.

**Supervision:** Timothy F Brady, John T Serences.

**Validation:** Anna Shafer-Skelton.

**Visualization:** Anna Shafer-Skelton.

**Writing – original draft:** Anna Shafer-Skelton.

**Writing – review & editing:** Anna Shafer-Skelton, Timothy F Brady, John T Serences.

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
