## [Decision Letter · Decision Letter 0]

26 Aug 2024

Dear Dr. Shafer-Skelton,

Thank you very much for submitting your manuscript "A 2D Gabor-wavelet baseline model out-performs a 3D surface model in scene-responsive cortex" for consideration at PLOS Computational Biology.

As with all papers reviewed by the journal, your manuscript was reviewed by members of the editorial board and by several independent reviewers. In light of the reviews (below this email), we would like to invite the resubmission of a significantly-revised version that takes into account the reviewers' comments.

We cannot make any decision about publication until we have seen the revised manuscript and your response to the reviewers' comments. Your revised manuscript is also likely to be sent to reviewers for further evaluation.

Sincerely,

Tim Christian Kietzmann, Dr. rer. nat.

Academic Editor

PLOS Computational Biology

Andrea E. Martin

Section Editor

PLOS Computational Biology

Reviewer's Responses to Questions

**Comments to the Authors:**

Reviewer #1: This study presents an fMRI experiment which utilizes voxel-wise encoding models to disentangle the neural representations of 2D vs. 3D information in BOLD responses to scene images in scene-selective cortex in the human brain. The 2D model is a Gabor wavelet model and the 3D models are derived from ground-truth distance and surface maps in the Taskonomy dataset. Stimuli were chosen from the larger Taskonomy dataset by minimizing representational distances between the different model features. The authors explicitly indicate that the central message of the paper is that their results contradict prior work that claims more (unique) representation of 3D scene information in scene-selective cortex than 2D information, focusing strongly on one high-profile paper by Lescroart & Gallant (2019) who used movie stimuli and similar models (but also several other studies, as highlighted in Figure 1). They argue that the previous claims of 3D representation may be due to co-variation of 3D cues with motion, or the use of weaker 3D baseline than those employed here.

MAIN COMMENTS

The study is well designed – the use of the Taskonomy images is clever given the availability of 3D ground truth data in those static images, and the fMRI experiment is highly powered, with carefully considered tasks. The model fitting procedure and ensuing analyses appear solid, and many of the points made by the authors are fair and will I think be important to communicate the field of scene perception research.

Stylistically, I am a bit unsure though about whether the strong contrast that is drawn with the Lescroart study is completely fair given that the authors here opted to remove motion information entirely (by using static stimuli) rather than running a new study with movies with the same or similar stimuli and including a motion-baseline model – if in that case no unique variance for 3D information is found, it would be a very strong non-replication. I’m not suggesting the authors run this study, but it might be worth setting up the Introduction a bit differently, so that the reader isn’t ‘disappointed’ when they find that the current study in fact didn’t use video (as I was when I arrived at page 7 “In the present study, rather than attempt to expand a baseline model for the complexities of video stimuli, we instead removed…”). The Introduction seems set up to present that.

The Results section is also oddly short. It is clear the authors want to make a punchy point and put all the ‘details’ in supplement, but I don’t see why that’s necessary for this journal as there are no page restrictions to report the analyses in the full text. It also doesn’t allow the reader to see a bit more of this potentially very interesting dataset without making extra effort to download the supplement. (NB The Supplementary figures appear to be screenshots of Matlab figure windows, some of which are not well readable (e.g. S3). I do not think this should be considered acceptable for publication, especially not since important additional analyses are presented in the Supplement). Also, the sole Results figure (Fig 3) is just bar plots: for a more comprehensive comparison with Lescroart & Gallant (2019), it would be good to include a visualization of encoding model performance on flatmaps along with ROI definitions (see also next point).

Although the authors clearly already thought deeply about their distinctive results from Lescroart & Gallant (2019), and did several additional analyses in an attempt to rule out alternative explanations, I think there may be one additional factor that could potential play a role, namely the ROI definitions: in Lescroart et al. any voxels in PPA, OPA and RSC that overlapped with retinotopic maps were excluded (I think this is the general custom in the Gallant lab). Although here the authors test for selecting subset of voxels, I don’t think the analyses included a systematic removal of only those voxels that overlapped with the same retinotopic maps as in L&G. Here, the authors also use mutually exclusive ROIs, but mapped out only V1-V3, while L&G included higher-order maps such as V4, V3AB, LO and V7. It’s perhaps not very likely that any of these will overlap with PPA (which does overlap with retinotopic maps anterior to V4, see Arcaro et al., 2009) but definitely some will overlap with OPA (see Silson et al., 2016). In other words, it seems pertinent to run a control analysis where OPA/PPA are defined without any retinotopic overlap similarly to Lescroart and colleagues to exclude an explanation of the difference in results between the studies based on the inclusion of retinotopic voxels.

MINOR COMMENTS

Order of presentation: Some of the Results sections could not be followed well without reading the Methods first; it would help readability to present Methods first, or to provide more context on the Results, e.g. to preface some more of the methods in Intro or start of the results. For example, from just Fig 2 we can’t tell if the method is RSA or voxelwise encoding or something else. Same for the selection of stimuli, it is not obvious from the caption of Fig 2 between which models the RDM correlation was explicitly minimized (from the Methods, it becomes clear it was never 2D vs 3D but that just turned out to be so as a side-effect of minimizing 3D global vs. 3D quadrant? Isn't that interesting in itself actually?). The Results section on voxel selection (page 12) it is unclear what ‘our pre-determined voxel counts’ refers to (without reading Methods).

Discussion P18: ‘..local affordance (arguably 3D); Bonner & Epstein 2017 never explicitly argued that their path drawings implicate a 3D representation, I think? So it would be good to substantiate this argument a bit more.

Methods Page 26: It would be good to say a bit more about how the ground-truth for the Taskonomy image maps were obtained to construct their training data. After all it is very important for the conclusions of this study that the 3D model is in fact accurate in representing the 3D structure of these (real-world) images.

Methods Page 28: The acronym ‘VOI’ is introduced here without explanation, is this the same as ROI or something different?

Figure S2: what about dot task only performance? Was there perhaps an overall reduction in correlations then? No difference in terms of model explained variances?

LANGUAGE/TYPOS

P8 ‘While we also manipulated… there seems to be an ‘of’ lacking here in this sentence (the spatial extent of?)

There are in-line bold headers/summary statements in the beginning of the Results (page 11) and Discussion, that do not always fit neatly in the flow of the text.

Author names in citation 20 (Groen et al., 2016) is not correctly formatted.

References:

Silson, E.H., Groen, I.I.A., Kravitz, D.J., and Baker, C.I. (2016). Evaluating the correspondence between face-, scene-, and object-selectivity and retinotopic organization within lateral occipitotemporal cortex. J. Vis. 16, 14.

Arcaro, M.J., McMains, S.A., Singer, B.D. and Kastner, S. (2009). Retinotopic organization of human ventral visual cortex. J Neurosci 29(34), 10638-10652.

Reviewer #2: This paper by Shafer-Skelton and colleagues, presents a body of fMRI data which nicely complement the existing literature on how 3D scenes are represented by the responses of ventral visual cortex. Their main conclusion is that previous reports of responses capturing 3D scene structure may not have adequately ruled out an alternative based on lower-level visual properties, providing a timely contribution to this research question. A nice feature of this paper is that the authors checked that their main findings were robust across different methodological decisions, and the specifics of the models used.

The paper is well written, the data are carefully presented, and the main claims are supported by the data.

My only substantial point is that the results would be more convincing if there were a model that better accounted for responses in the higher level areas (OPA, PPA, RSC) than the early visual areas (V1-3), and/or if the data demonstrated any qualitative differences between ROIs. In the current data, as presented in Figure 3, the overall impression is that signals are much weaker in the higher level areas, and that there aren’t any substantial differences in feature response that have been identified with this dataset. This doesn’t undermine the main conclusions, since they are based on the relative performance of the 3D vs 2D models within each ROI. The authors note that ‘magnitudes across ROIs are not interpretable, but patterns within ROIs are’ – I understand the claim here, but the magnitudes might reflect SNR. Even if there are no models that perform better in accounting for responses in higher-level ROIs than early ones, it would be helpful to demonstrate that there is enough signal to detect selectivity differences between the ROIs. One prediction that could be useful here is that higher level ventral areas tend to show less dependence on spatial frequency (e.g. https://doi.org/10.1371/journal.pcbi.1004719
https://doi.org/10.1016/j.neuroimage.2020.116780). If the current fMRI data showed a similar trend it would at least demonstrate that the dataset have sufficient SNR to detect differences between ROIs.

Minor queries:

-Top of page 8: I think I’m missing something here, but it sounds as though they’re saying that the baseline model they’ll include is similar to the Gabor model, but now encompasses SF and orientation – in what way would a Gabor model not include SF and orientation?

-The final sentence of introduction seemed to come a bit out of left field?

-Methods: bottom of page 25: ‘one of 4 spatial frequencies’, but 5 values listed? Is the ‘0’ spatial frequency no included at each orientation?

-Some of the figures in the supplementary material are very hard to read (low res, small). It would be good if any materials that are included are presented at adequate size and resolution.

**Have the authors made all data and (if applicable) computational code underlying the findings in their manuscript fully available?**

Reviewer #1: **No:** The github link provided is not active so it's not possible to check if the all data aspects are available

Reviewer #2: **No:** I followed the GitHub link but got a 404 error

PLOS authors have the option to publish the peer review history of their article (what does this mean? ). If published, this will include your full peer review and any attached files.

**Do you want your identity to be public for this peer review?** For information about this choice, including consent withdrawal, please see our Privacy Policy .

Reviewer #1: **Yes:** Iris Groen

Reviewer #2: No
---

## [Decision Letter · Decision Letter 1]

5 Jun 2025

A 2D Gabor-wavelet baseline model out-performs a 3D surface model in scene-responsive cortex

PLOS Computational Biology

Dear Dr. Shafer-Skelton,

Thank you for submitting your manuscript to PLOS Computational Biology. After careful consideration, we feel that it has merit but does not fully meet PLOS Computational Biology's publication criteria as it currently stands. Therefore, we invite you to submit a revised version of the manuscript that addresses the points raised during the review process.

Please submit your revised manuscript within 60 days Aug 05 2025 11:59PM. If you will need more time than this to complete your revisions, please reply to this message or contact the journal office at ploscompbiol@plos.org. Please include the following items when submitting your revised manuscript:

We look forward to receiving your revised manuscript.

Kind regards,

Tim Christian Kietzmann, Dr. rer. nat.

Academic Editor

PLOS Computational Biology

Andrea E. Martin

Section Editor

PLOS Computational Biology

**Reviewers' comments:**

Reviewer's Responses to Questions

**Comments to the Authors:**

Reviewer #1: I went through the revised manuscript and the author’s response, and I think the paper has improved in multiple aspects. Nevertheless, I still have a few concerns that I believe need to be addressed:

- The Introduction is now much better in terms of balance regarding the prior literature, no longer creating the impression the authors are attempting to refute Lescroart & Gallant [13] through the same setup (video) with motion control. It seems like the Abstract however has not been updated to reflect this change in framing. Especially the sentence “this raises the question…” is, in my opinion, too speculative for an Abstract – after all, the current manuscript does not directly address the low-level motion control for that study. I suggest rephrasing the Abstract to focus more on the contributions of the current work and less on the implications regarding validity of earlier claims (similar to the Author Summary).

- It seems the authors have not followed my suggestion to include an additional paragraph at the start of the results that summarizes the methods to better prepare the reader. I still think it would greatly benefit the accessibility of the paper to do so, i.e. to briefly summarize the fMRI experiment setup, explain that you had two behavioral tasks, etc., walking the reader through all panels of Fig 2 (which is currently skipped in text in the Intro/Results), and explain that the fMRI analysis consists of voxelwise encoding and variance partitioning.

- Results: “Note that none of our results are normalized to a noise ceiling, so we cannot compare absolute magnitudes with noise-ceiling-normalized results.” What prohibits you from computing a noise ceiling and plotting the results in that way, facilitating the detailed comparison?

- For the surface flatmaps: Analogous to the ROI analyses, in addition to the unique partitions, please also show full/raw correlations for the 2D and 3D models for each participant, so those can be compared to prior literature. One citation that the authors missed but that seems particularly relevant in this context is Dwivedi et al., PCB 2021, who report differential mapping of DNNs trained on the Taskonomy dataset on 2D vs 3D tasks in dorsal vs. ventral ROIs.

- Upon reading the paper again, I realized the study design seem geared towards testing a different hypothesis; perhaps regarding the spatial locality of task-relevant 3D information? It would be good to explicitly mention this as now there is no motivation provided for several aspects of the experimental design.

- Discussion; I appreciate the newly added sections on 23-26, aiming to clarify the difficulty of disentangling ‘low-level’ vs. ‘high-level’ features in scenes. However, this section makes the Discussion section quite lengthy, and in the end I’m not quite sure what the authors want to say; are they arguing against using variance partitioning as a method (cf, page 25/26 “Without a more thorough understanding…”)? The current Discussion both praises and criticizes the use of such methods on naturalistic stimuli (in contrast to controlled stimuli) but doesn’t clearly outline what should then be improved (note that the reference to task manipulations in the last paragraph seems quite out of place, given that no task effect was found on the encoding model performances here). How do they envision we obtain such more thorough model understanding – do we need a different paradigm for this? I suggest shortening and consolidating these sections around key points the authors want to make.

Minor comments:

- Small note on Introduction: It wasn’t immediately clear to me what ‘To this end’ at the start of the second paragraph refers to: to ‘disambiguating representations’, or ‘studying the content of scene information’? It would be good to explicitly spell out the goal here.

- Results Page 12 “We found similar magnitudes of cross-validated prediction performance” – the way this is phrased made me initially (I hope erroneously) think that the main results reported in Fig 3 are not cross-validated. I think the authors are trying to emphasize the ‘magnitudes’ part, but since the sentence before it does not mention ‘cross-validated’, it seems like that part is new. Also, it would be good to explicitly say ‘similar magnitudes … as in reference [13]’.

- Results: In the pdf I received to review, the text-embedded Figures 4 and S2 were of poor quality, with blurred text and completely illegible axes labels for panels 4B-C. Also, the separately uploaded .tiff files looked blurry in the pdf (but those other figures then again looked OK in-text). Please ensure sufficient figure quality by exporting figures with high resolution, and evaluating converted pdf quality.

- In the revised Results section, the panels of Figure 4 are discussed in reverse order. Please either rearrange the figure or change the text to ensure the text introduces each panel in the correct alphabetical order.

- Figure 5 is currently not discussed or described in text.

- The quality of the flatmap figures can be improved. Because of the use of black backgrounds and showing the Gabor model performance in dark blue, vertices with significant performance are hard to discern – it might be much clearer when using a bright color. The ROI boundaries have varying colors/color intensities that are not explained in a legend (initially I was mistaking the purple ROI boundaries in Fig 5 for R2 by the Gabor model, as this is how it is indicated in the bar plots), and the text in Figure 5 is again somewhat blurry (curiously, Figure S3 looks much better). The error bar labels for the flatmaps show overlapping text labels and are of low resolution – it looks like it was made via a screenshot from a GUI. It may help to edit or remake such figure elements in a separate figure editing program (e.g. Illustrator or Inkscape) to ensure good legibility.

- Figure 5 and S3: All participants seem to show some unique 3D variance anterior of RSC. Do the authors want to comment on this? Is this noise, or potentially meaningful?

- Discussion page 24 on baselines “This is especially true when…” but [13] still found a unique contribution of 3D despite picking the ‘best baseline’ in this way, so I don’t see how this supports the point regarding the flexibility of baseline models here? Please clarify.

Reviewer #2: This revised manuscript includes a number of improvements over the original submission, primarily in how clearly the data are presented, with an improved motivation for the study, and with further elaboration on some points in the discussion.

However, I don’t feel that the authors have really addressed the only major comment I raised in my original review, namely that they fail to find evidence of any differences between ROIs. The main result in this study is essentially a null effect (no difference across ROIs in the relative performance of 2D vs 3D models), which argues against conclusions of previous studies, and the methodological approach is relatively complex, making it harder to intuitively understand its limitations. Given these factors, it would be reassuring if the authors could demonstrate that their data and modelling approach can be used replicate other known differences between regions, such as their dependence on spatial frequency. Being able to show that they can reveal differences across ROIs for coding of an alternate stimulus dimension would show be a helpful way to demonstrate that the ‘null result’ of no difference across ROIs is less like to be attributed to the data have insufficient signal to noise in higher level areas, or the modelling approach not being sensitive to differences between ROIs for some reason.

Minor points

- some figures remain too low resolution to read, such as Figure 4

- the new Figure 5 is not referred to or discussed in the body of the Results text.

**Have the authors made all data and (if applicable) computational code underlying the findings in their manuscript fully available?**

Reviewer #1: Yes

Reviewer #2: Yes

PLOS authors have the option to publish the peer review history of their article (what does this mean? ). If published, this will include your full peer review and any attached files.

**Do you want your identity to be public for this peer review?** For information about this choice, including consent withdrawal, please see our Privacy Policy .

Reviewer #1: No

Reviewer #2: No

**Figure resubmission:**

**Reproducibility:**



---

## [Decision Letter · Decision Letter 2]

5 Jan 2026

Dear Dr. Shafer-Skelton,

We are pleased to inform you that your manuscript 'A 2D Gabor-wavelet model out-performs 3D surface model in scene-responsive cortex' has been provisionally accepted for publication in PLOS Computational Biology.

Before your manuscript can be formally accepted you will need to complete some formatting changes, which you will receive in a follow up email. Please also note a few minor details described by reviewer 1 that still need addressing.

A member of our team will be in touch with a set of requests.

Best regards,

Tim Christian Kietzmann, Dr. rer. nat.

Academic Editor

PLOS Computational Biology

Andrea E. Martin

Section Editor

PLOS Computational Biology

Reviewer's Responses to Questions

**Comments to the Authors:**

Reviewer #1: The revised paper is much improved, and I have no further recommendations for the content or analyses.

A few small issues remain with presentation, however:

- Page 12 refers to Figure 2B as showing ‘three encoding models’ , but the panel only shows the 3D models. It would be helpful to include a depiction of the 2D model also (I would recommend to remove panel A, which is just text and partly redundant with panel D, and instead used the top row to clearly illustrate the three different feature models used in this study).

- Figure 6 is not referenced/described in the Results text.

- Figure 7ACE are the same as 3ABC, hence redundant.

- Supplementary Tables S2A-C exists of what appear to be command window/terminal figures instead of actual in-text formatted tables, using unexplained abbreviations as column names.

Reviewer #2: The authors have address all my remaining concerns.

**Have the authors made all data and (if applicable) computational code underlying the findings in their manuscript fully available?**

Reviewer #1: Yes

Reviewer #2: **No:** I'm unclear on whether the data included in the github repository satisfies this requirement: e.g. there's no individual data, so no equivalent of 'the data points behind means'.

PLOS authors have the option to publish the peer review history of their article (what does this mean? ). If published, this will include your full peer review and any attached files.

**Do you want your identity to be public for this peer review?** For information about this choice, including consent withdrawal, please see our Privacy Policy .

Reviewer #1: No

Reviewer #2: No

---

## [Editor Report · Acceptance letter]

PCOMPBIOL-D-24-00406R2

A 2D Gabor-wavelet model out-performs a 3D surface model in scene-responsive cortex

Dear Dr Shafer-Skelton,

I am pleased to inform you that your manuscript has been formally accepted for publication in PLOS Computational Biology. Your manuscript is now with our production department and you will be notified of the publication date in due course.

With kind regards,

Swetha Kaliappan
